# Calcium signaling from damaged lysosomes induces cytoprotective stress granules

Jacob Duran[1,2,8], Jay E Salinas[1,2,8], Rui ping Wheaton [1,2,8], Suttinee Poolsup [1,2], Lee Allers[2,3], Monica Rosas-Lemus [2,3], Li Chen[2,3], Qiuying Cheng[1], Jing Pu [3], Michelle Salemi[4], Brett Phinney[4], Pavel Ivanov[5], Alf Håkon Lystad[6], Kiran Bhaskar [3,7], Jaya Rajaiya[3], Douglas J Perkins[1] & Jingyue Jia [1,2✉]

## Abstract

**Lysosomal damage induces stress granule (SG) formation. However, the importance of SGs in determining cell fate and the precise mechanisms that mediate SG formation in response to lysosomal damage remain unclear. Here, we describe a novel calcium-dependent pathway controlling SG formation, which promotes cell survival during lysosomal damage. Mechanistically, the calcium-activated protein ALIX transduces lysosomal damage signals to SG formation by controlling eIF2α phosphorylation after sensing calcium leakage. ALIX enhances eIF2α phosphorylation by promoting the association between PKR and its activator PACT, with galectin-3 inhibiting this interaction; these regulatory events occur on damaged lysosomes. We further find that SG formation plays a crucial role in promoting cell survival upon lysosomal damage caused by factors such as SARS-CoV-2[ORF3a], adenovirus, malarial pigment, proteopathic tau, or environmental hazards. Collectively, these data provide insights into the mechanism of SG formation upon lysosomal damage and implicate it in diseases associated with damaged lysosomes and SGs.**

**Keywords** Lysosomal Damage; Stress Granules; Calcium-dependent Pathway; ALG2-ALIX; PACT-PKR-eIF2α
**Subject Categories** Microbiology, Virology & Host Pathogen Interaction; Organelles

## Introduction

Lysosomes are acidic hydrolase-rich membrane-bound organelles that play a vital role in cellular degradation and signaling (Ballabio and Bonifacino, 2020; Lamming et al, 2019; Lawrence and Zoncu, 2019; Yang et al, 2021). Damage to lysosomes can be triggered by numerous physiological and pathological conditions (Nakamura et al, 2021; Papadopoulos et al, 2017; Yang and Tan, 2023). These include microbial pathogens (Ghosh et al, 2020; Montespan et al, 2017; Thurston et al, 2012), environmental pollutants (Hornung et al, 2008; Mossman and Churg, 1998; J. Wang et al, 2017), toxic protein aggregates (Flavin et al, 2017; Papadopoulos et al, 2017), endogenous crystals (Hui et al, 2012; Maejima et al, 2013), and many lysosomotropic drugs (Marceau et al, 2012; Pisonero-Vaquero and Medina, 2017). These agents, along with various others, damage lysosomes, leading to the leakage of acidic contents and the disruption of cellular functions, thereby threatening cell survival (Patra et al, 2023; Saftig and Puertollano, 2021; Wang et al, 2018). Lysosomal damage is strongly linked to various human diseases, e.g., cancer, infectious, and neurodegenerative diseases (Amaral et al, 2023; Ballabio and Bonifacino, 2020; Bonam et al, 2019; Fehrenbacher et al, 2005). Although lysosomal damage is of physiological importance and pathological relevance, understanding of how cells respond to this damage remains largely unknown (Papadopoulos and Meyer, 2017).

Cells can detect lysosomal damage through several mechanisms, including the identification of calcium leakage or the exposure of luminal glycan (Aits et al, 2015; Radulovic et al, 2018; Skowyra et al, 2018). Minorly damaged lysosomes can be repaired through multiple cellular systems, including annexins (Ebstrup et al, 2023; Yim et al, 2022), sphingomyelin turnover (Niekamp et al, 2022), microautophagy (Ogura et al, 2023), ER-lysosome lipid transfer (Radulovic et al, 2022; Tan and Finkel, 2022) as well as ESCRT (the endosomal sorting complexes required for transport) machinery (Radulovic et al, 2018; Skowyra et al, 2018). Notably, the protein ALIX (ALG-2-Interacting Protein X), a key ESCRT component, can detect lysosomal damage by sensing calcium release, a function it performs alongside its partner, ALG2 (Apoptosis-Linked Gene-2) (Chen et al, 2024; Maki et al, 2016; Sun et al, 2015). Upon detecting such damage, ALIX facilitates the recruitment of other ESCRT components to the site of damage for repair (Chen et al, 2024; Radulovic et al, 2018; Skowyra et al, 2018). Severely damaged lysosomes can be removed by selective autophagy (Chauhan et al,

[1]Center for Global Health, Department of Internal Medicine, University of New Mexico Health Sciences Center, Albuquerque, NM 87106, USA.[2]Autophagy, Inflammation and Metabolism Center of Biochemical Research Excellence, Albuquerque, NM 87106, USA.[3]Department of Molecular Genetics and Microbiology, University of New Mexico Health Sciences Center, Albuquerque, NM 87106, USA.[4]Proteomics Core Facility, University of California Davis Genome Center, University of California, Davis, CA 95616, USA.[5]Department of Medicine, Brigham and Women's Hospital and Harvard Medical School; HMS Initiative for RNA Medicine, Boston, MA 02115, USA.[6]Centre for Cancer Cell Reprogramming, Faculty of Medicine, University of Oslo; Department of Molecular Cell Biology, Institute for Cancer Research, Oslo University Hospital, Oslo, Norway.[7]Department of Neurology, University of New Mexico Health Sciences Center, Albuquerque, NM 87106, USA.[8]These authors contributed equally: Jacob Duran, Jay E Salinas, Rui ping Wheaton. ✉E-mail: JJia@salud.unm.edu

2016; Maejima et al, 2013), noncanonical autophagy (Boyle et al, 2023; Kaur et al, 2023), or lysosomal exocytosis (Domingues et al, 2024; Wang et al, 2023). Master regulators mTORC1 (mechanistic target of rapamycin complex 1) and AMPK (AMP-activated protein kinase), located on lysosomes (Sancak et al, 2010; Zhang et al, 2014), are finely tuned to respond to lysosomal damage, subsequently activating downstream processes e.g., autophagy and lysosomal biogenesis (Jia et al, 2018; Jia et al, 2020a, 2020b; Jia et al, 2020c). These mechanisms collectively safeguard lysosomal quality, maintaining cellular homeostasis (Jia et al, 2020d).

Recently, we reported that lysosomal damage induces the formation of stress granules (SGs) (Jia et al, 2022; Jia et al, 2023). SGs are membrane-less organelles identified as ribonucleoprotein condensates that are believed to serve as protective responses in cells under adverse conditions (Ivanov et al, 2019; McCormick and Khaperskyy, 2017; Riggs et al, 2020). Consequently, dysfunctional SGs have been implicated in various human diseases e.g., neurodegenerative and infectious diseases(Advani and Ivanov, 2020; Protter and Parker, 2016; Wang et al, 2020). SG formation is triggered by specific kinases, such as PKR (Protein Kinase R), that sense various stress stimuli, leading to the phosphorylation of eIF2α (eukaryotic translation initiation factor 2) (Kedersha et al, 1999; Srivastava et al, 1998). Phosphorylated eIF2α (p-eIF2α) halts global translation, resulting in the accumulation of untranslated mRNA (Jackson et al, 2010). Simultaneously, it promotes the selective expression of stress response proteins, a process known as the integrated stress response (Costa-Mattioli and Walter, 2020; Pakos-Zebrucka et al, 2016). SG formation can also occur through mTORC1-mediated translational shutdown, independent of p-eIF2α (Emara et al, 2012; Fujimura et al, 2012; McCormick and Khaperskyy, 2017). RNA-binding proteins G3BP1/2 (GAP SH3 Domain-Binding Protein 1/2) detect untranslated mRNA and collectively initiate SG formation through an RNA-protein network, driven by liquid-liquid phase separation (Hyman et al, 2014; Ivanov et al, 2019).

Despite the extensive knowledge of SG composition and dynamics, an understanding of the functional consequences of SG formation remains limited (Riggs et al, 2020). SG formation has often been investigated under non-physiological conditions such as arsenic stress or heat shock (Jain et al, 2016; Sidrauski et al, 2015; Turakhiya et al, 2018; Verma et al, 2021; Yang et al, 2020). Notably, our study (Jia et al, 2022) which originally revealed lysosomal damage as a critical internal physiological trigger for SGs, underscores the need to better understand the nature of SG formation in disease contexts. In addition, this new connection between damaged lysosomes and SGs provides a novel perspective on the interaction between membrane-bound and membrane-less organelles (Zhao and Zhang, 2020). For example, recent research suggests that SGs have the ability to plug and stabilize damaged lysosomes (Bussi et al, 2023). However, the precise regulation of SG formation in response to lysosomal damage and its consequential impact on cell fate remains largely unexplored.

In this study, we employed unbiased approaches to investigate how lysosomal damage signals are transduced to induce SG formation and to elucidate the cytoprotective role of SG formation in promoting cell survival against lysosomal damage. Our findings revealed a novel function of ALIX, which senses calcium release from damaged lysosomes, in controlling the phosphorylation of eIF2α through PKR and its activator on damaged lysosomes, thereby initiating SG formation. This process is critical for cell survival in response to lysosomal damage caused by microbiological, pathological, and environmental agents

including SARS-CoV-2^ORF3a, adenovirus, Malaria hemozoin, proteopathic tau and silica. In conclusion, our study uncovers a calcium-dependent signaling mechanism that transmits lysosomal damage signals to induce SG formation and reveals the cytoprotective role of SG formation in response to lysosomal damage caused by diverse stresses.

# Results

## Stress granule formation promotes cell survival in response to lysosomal damage

How does SG formation affect cell fate during lysosomal damage? We utilized SG deficient U2OS cells (human osteosarcoma epithelial cell line) genetically lacking both G3BP1 and G3BP2 (ΔΔG3BP1/2) (Kedersha et al, 2016), which are essential factors for SG formation (Guillén-Boixet et al, 2020; Kedersha et al, 2016; Yang et al, 2020) (Fig. EV1A). We quantified the number of SGs using the canonical SG marker polyA RNA (Ivanov et al, 2019) via high-content microscopy (HCM) and verified the depletion of SG formation in ΔΔG3BP1/2 cells when exposed to the lysosome-specific damaging agent L-leucyl-L-leucine methyl ester (LLOMe) (Jia et al, 2022; Tan and Finkel, 2022; Thiele and Lipsky, 1990) (Fig. EV1B(i)). We also found that depleting G3BP1 and G3BP2 does not impact lysosomal biogenesis, as indicated by the expression and puncta formation of the lysosomal integrated protein LAMP2 (Figs. EV1A, 1B(ii)). A propidium iodide (PI) uptake assay measuring plasma membrane integrity (Crowley et al, 2016; Liu et al, 2023) was adapted to quantify cell survival during lysosomal damage using HCM. We found significant cell death upon LLOMe treatment in ΔΔG3BP1/2 cells compared to wild type (WT) U2OS cells (Fig. 1A). This was additionally confirmed by using a lactate dehydrogenase (LDH) release assay measuring non-specific leak from cells (Chan et al, 2013; Kumar et al, 2018) (Fig. 1B). Further, we pharmacologically blocked SG assembly through the use of cyclohex-imide which freezes ribosomes on translating mRNAs and reduces the accumulation of free untranslated mRNA (Freibaum et al, 2021; Kedersha et al, 2000). Consistent with previous reports (Bussi et al, 2023; Jia et al, 2022), cycloheximide treatment inhibited SG formation in U2OS cells, as evidenced by the absence of G3BP1 puncta following LLOMe treatment (Fig. EV1C). This suppression of SG formation led to reduced cell survival, as indicated by increased LDH release in the face of lysosomal damage (Fig. EV1D). Previously we reported that LLOMe treatment induced phosphorylation of eIF2α (Jia et al, 2022), a critical signal for SG formation (Ivanov et al, 2019; Kedersha et al, 2000). The small molecule ISRIB (integrated stress response inhibitor) can also act as an SG inhibitor, effectively counteracting the downstream effects of eIF2α phosphorylation, such as ATF4 (Activating transcription factor 4) expression (Rabouw et al, 2019; Sidrauski et al, 2015). We prevented SG formation using ISRIB upon lysosomal damage (Fig. EV1E) and observed a corresponding reduction in ATF4 expression levels in THP-1 cells (the human monocytic cell line) (Fig. EV1F). The prevention of SG formation by ISRIB also caused a decrease in cell survival in THP-1 cells (Fig. EV1G). Furthermore, the cell death effect during lysosomal damage caused by the loss of SG formation can be rescued. This is evidenced by the reduced cell death, as measured by the PI uptake assay, when G3BP1 and G3BP2 were overexpressed in ΔΔG3BP1/2 cells (Fig. EV1H).

The protective effects of SG formation in response to lysosomal damage were also observed in primary cells using human peripheral

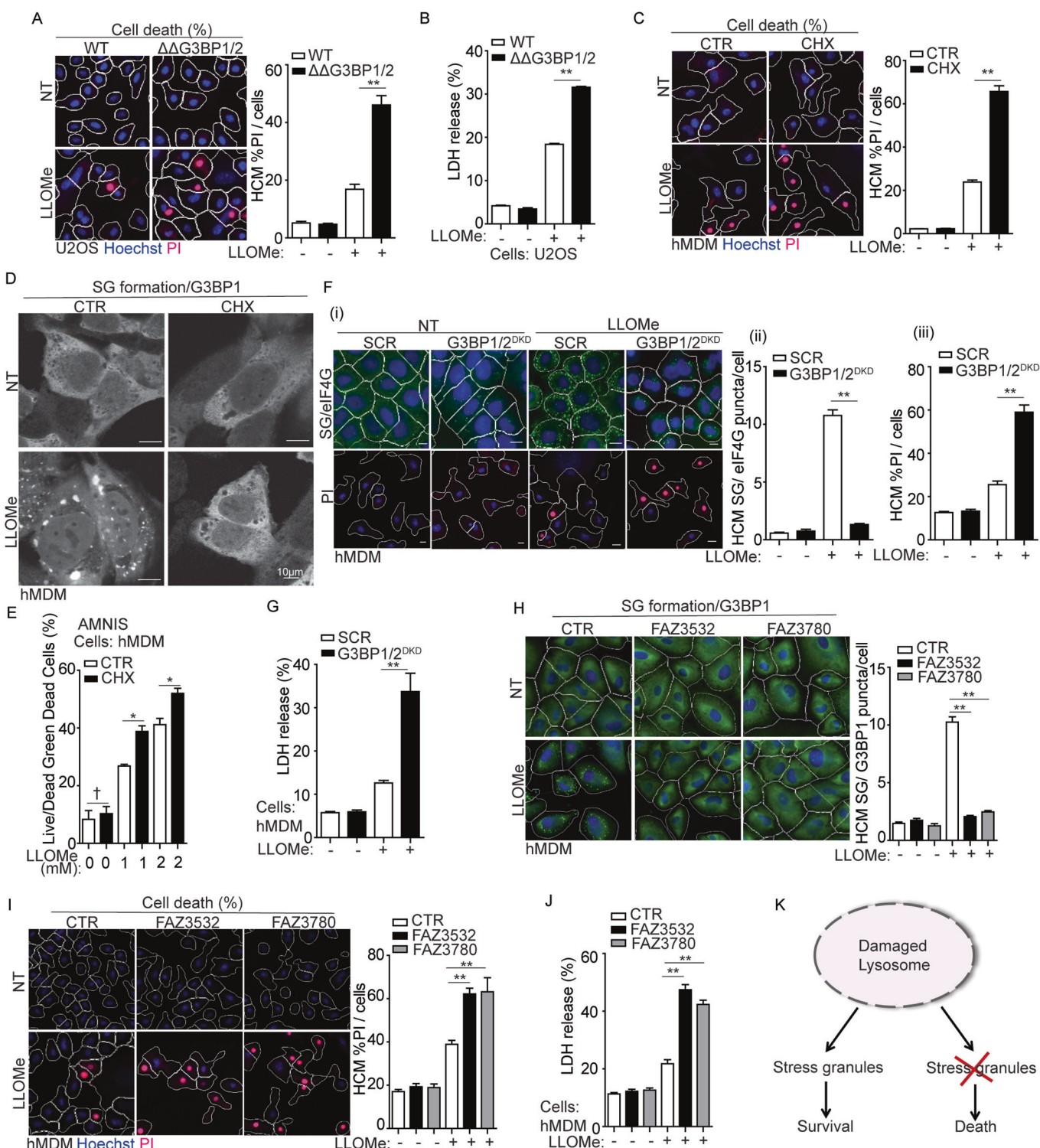

blood monocyte-derived macrophages (hMDM). This includes that the significant increase in cell death during LLOMe treatment, as quantified by the PI uptake assay when SG formation was inhibited by cycloheximide in hMDM (Fig. 1C,D). This was further confirmed by measuring the viability of live hMDM (without the fixation) using an AMNIS imaging flow cytometer (Fig. 1E).

Knockdown of both G3BP1 and G3BP2 in hMDM (G3BP1/2$^{DKD}$) resulted in a reduction of SG formation as evaluated by a key SG marker, eIF4G puncta, during LLOMe treatment (Figs. 1F (i, ii) and EV1I). Elevated cell death, as quantified by PI uptake assay (Fig. 1F (i, iii)) and the LDH release assay (Fig. 1G), was detected in G3BP1/2$^{DKD}$ in response to LLOMe treatment.

**Figure 1. Stress granule formation promotes cell survival in response to lysosomal damage.**

(A) Quantification by high-content microscopy (HCM) of cell death by a propidium iodide (PI) uptake assay in U2OS wild type (WT) and G3BP1&2 double knockout (ΔΔG3BP1/2) cells. Cells were treated with 2 mM LLOMe for 30 min, and then stained with propidium iodide (PI) (dead cells) and Hoechst-33342 (total cells). White masks, algorithm-defined cell boundaries (primary objects); red masks, computer-identified PI⁺ nuclei (target objects). (B) Cell death analysis of supernatants of U2OS WT and ΔΔG3BP1/2 cells by a LDH release assay. Cells were treated with 2 mM LLOMe for 30 min. (C) Quantification by HCM of cell death by a PI uptake assay in human peripheral blood monocyte-derived macrophages (hMDM). Cells were treated with 2 mM LLOMe in the presence or absence of 10 μg/ml cycloheximide (CHX) for 30 min, and then stained with PI (dead cells) and Hoechst-33342 (total cells). (D) Confocal microscopy analysis of G3BP1 (Alexa Fluor 488) in hMDM treated with 2 mM LLOMe with or without CHX for 30 min. Scale bar, 10 μm. (E) Quantification using AMNIS of cell death by Live/Dead™ stain kit in hMDM. Cells were treated with 2 mM LLOMe with or without CHX for 30 min, and then stained using Live/Dead™ stain kit (ThermoFisher). (F) Quantification by HCM of cell death by a PI uptake assay and SG formation by eIF4G in hMDM transfected with scrambled siRNA as control (SCR) or G3BP1 and G3BP2 siRNA for double knockdown (DKD). Cells were treated with 2 mM LLOMe for 30 min, and then stained with PI (dead cells), Hoechst-33342 (total cells) or eIF4G. (i) HCM images: white masks, algorithm-defined cell boundaries; green masks, computer-identified eIF4G puncta; red masks, computer-identified PI+ nuclei (target objects); (ii and iii) corresponding HCM quantification. Scale bar, 10 μm. (G) Cell death analysis of supernatants of hMDM transfected with either scrambled siRNA as control (SCR) or G3BP1 and G3BP2 siRNA for double knockdown (DKD) using a LDH release assay. Cells were treated with 2 mM LLOMe for 30 min. (H) Quantification by HCM of SG formation by G3BP1 in hMDM treated with 20 μM FAZ3532 or 20 μM FAZ3780 for 20 min, followed by exposure to 2 mM LLOMe for 30 min. Control cells were treated with DMSO. Green masks, computer-identified G3BP1 puncta. (I) Quantification by HCM of cell death by a PI uptake assay in hMDM treated with 20 μM FAZ3532 or 20 μM FAZ3780 for 20 min, followed by exposure to 2 mM LLOMe for 30 min. Control cells were treated with DMSO. Red masks, computer-identified PI+ nuclei. (J) Cell death analysis of supernatants of hMDM treated with 20 μM FAZ3532 or 20 μM FAZ3780 for 20 min, followed by exposure to 2 mM LLOMe for 30 min using a LDH release assay. Control cells were treated with DMSO. (K) Schematic summary of the findings in Fig. 1 and EV1. CTR, control; NT, untreated cells. Data, means ± SEM ($n=3$); HCM: $n \geq 3$ (each experiment: 500 valid primary objects/cells per well, ≥5 wells/sample). †$p \geq 0.05$ (not significant), *$p < 0.05$, **$p < 0.01$, ANOVA. See also Fig. EV1. Source data are available online for this figure.

To further validate the protective role of SG formation during lysosomal damage, we employed G3BP small-molecule inhibitors FAZ3532 and FAZ3780. These inhibitors bind to the dimerization domain of G3BP1/2, specifically disrupting the co-condensation of RNA, G3BP and SG network (Freibaum et al, 2024). First, we treated hMDM with these inhibitors and observed that they effectively inhibited SG formation induced by LLOMe individually (Fig. 1H). In addition, we found that FAZ3532/FAZ3780-induced SG deficiency significantly increased cell death upon lysosomal damage, as demonstrated by PI uptake assay (Fig. 1I) and LDH release assay (Fig. 1J). These data emphasize that SG assembly itself is necessary for cell survival during lysosomal damage. In summary, SG formation is a cytoprotective response to lysosomal damage (Fig. 1K).

## Stress granule formation is controlled by eIF2α pathway but not mTORC1 pathway during lysosomal damage

Considering the significance of SG formation during lysosomal damage, what mechanisms regulate SG formation in response to such damage? SG formation occurs as a consequence of protein translation arrest during cellular stress (Riggs et al, 2020; Youn et al, 2019). eIF2α phosphorylation and mTORC1 inactivation are two key upstream events that lead to protein translation arrest and subsequently trigger SG formation (Cotto and Morimoto, 1999; Emara et al, 2012; McCormick and Khaperskyy, 2017). Consistent with our earlier studies (Jia et al, 2018; Jia et al, 2022), we confirmed that LLOMe treatment induced eIF2α phosphorylation and mTORC1 inactivation (as assessed by the decreased phosphorylation of its substrates: 4EBP1 (Ser65), S6K (Thr389), ULK1 (Ser757), and TFEB (Ser142)), in a dose-dependent manner in U2OS cells (Fig. EV2A). To investigate the role of eIF2α and mTORC1 pathways in regulating SG formation upon lysosomal damage, we initially knocked down eIF2α in U2OS cells (eIF2α^KD) (Fig. 2A). This revealed that eIF2α is necessary for SG formation upon lysosomal damage, which was reflected by the depletion of SG formation in eIF2α^KD cells during LLOMe treatment (Fig. 2A). In addition, mTORC1 activity in eIF2α^KD cells was examined by detecting the phosphorylation of its substrates 4EBP1 (Ser65), S6K

(Thr389), ULK1 (Ser757) and TFEB (Ser142), revealing that mTORC1 inactivation was not affected by eIF2α depletion upon lysosomal damage (Fig. 2B). This indicates that eIF2α phosphorylation and mTORC1 inactivation are two uncoupled events during lysosomal damage. This was further confirmed by the lack of change in eIF2α phosphorylation upon lysosomal damage in cells expressing constitutively active RagB^Q99L, which keeps mTORC1 in an active state (Abu-Remaileh et al, 2017; Sancak et al, 2010) (Fig. 2C). In addition, SG formation was not affected in cells expressing RagB^Q99L in response to lysosomal damage (Fig. 2D). This uncoupled relationship between eIF2α phosphorylation and mTORC1 inactivation in SG formation is also reflected in various cellular stress conditions, including amino acid starvation and arsenic stress (Fig. EV2B,C). We found that amino acid starvation resulted in mTORC1 inactivation (assessed by mTOR dissociation from the lysosomes (Abu-Remaileh et al, 2017; Jia et al, 2022) but not eIF2α phosphorylation or SG formation as in previous reports (Prentzell et al, 2021; Wang and Proud, 2008) (Fig. EV2B,C). In contrast, arsenic stress led to eIF2α phosphorylation and SG formation while activating mTORC1 activity, consistent with earlier studies (Chen and Costa, 2018; Prentzell et al, 2021; Thedieck et al, 2013) (Fig. EV2B,C). The key role of eIF2α phosphorylation in SG formation during lysosomal damage was further demonstrated by the ability to complement eIF2α WT but not its phosphorylation site mutant (eIF2α S51A) (Kedersha et al, 1999) in eIF2α^KD cells to restore SG formation (Fig. 2E). In summary, eIF2α phosphorylation is a major upstream event for SG formation in response to lysosomal damage (Fig. 2F).

## Proteomics proximity analysis of eIF2α upon lysosomal damage reveals that its phosphorylation is driven by PKR and PACT

To further investigate the mechanisms that trigger eIF2α phosphorylation in response to lysosomal damage, we conducted a dynamic proteomic analysis using proximity biotinylation with APEX2-eIF2α fusion. First, we tested the kinetics of eIF2α phosphorylation upon LLOMe treatment in HEK293T cells

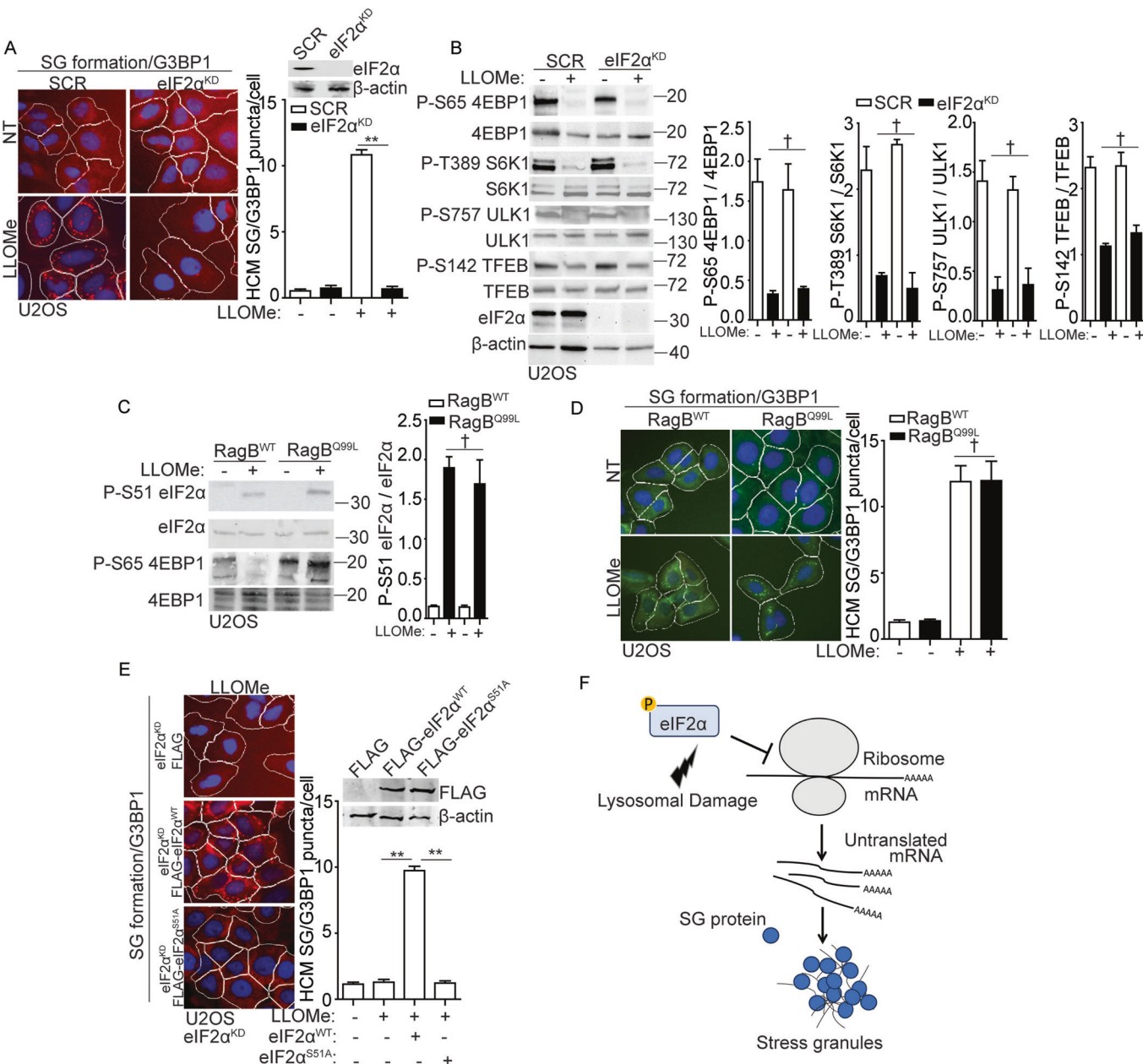

**Figure 2. Stress granule formation is controlled by eIF2α pathway but not mTORC1 pathway during lysosomal damage.**

(**A**) Quantification by HCM of G3BP1 puncta in U2OS cells transfected with either scrambled siRNA as control (SCR) or eIF2α siRNA for knockdown (eIF2α$^{KD}$). Cells were treated with 2 mM LLOMe for 30 min. White masks, algorithm-defined cell boundaries; red masks, computer-identified G3BP1 puncta. (**B**) Immunoblot analysis of mTORC1 activity by phosphorylation of 4EBP1 (Ser65), S6K (Thr389), ULK1 (Ser757), and TFEB (Ser142) in U2OS cells transfected with either scrambled siRNA as control (SCR) or eIF2α siRNA for knockdown (eIF2α$^{KD}$). Cells were treated with 2 mM LLOMe for 30 min. Quantification is based on three independent experiments. (**C**) Immunoblot analysis of phosphorylation of eIF2α (S51) in U2OS cells overexpressing wild-type RagB (RagB$^{WT}$) or constitutively active RagB mutant (RagB$^{Q99L}$) treated with 2 mM LLOMe for 30 min. Quantification is based on three independent experiments. (**D**) Quantification by HCM of G3BP1 puncta in U2OS cells overexpressing wild-type RagB (RagB$^{WT}$) or constitutively active RagB mutant (RagB$^{Q99L}$). Cells were treated with 2 mM LLOMe for 30 min. White masks, algorithm-defined cell boundaries; green masks, computer-identified G3BP1 puncta. (**E**) Quantification by HCM of G3BP1 puncta in eIF2α knockdown (eIF2α$^{KD}$) U2OS cells transfected with FLAG, FLAG- eIF2α$^{WT}$ or FLAG- eIF2α$^{S51A}$. Cells were treated with 2 Mm LLOMe for 30 min. White masks, algorithm-defined cell boundaries; red masks, computer-identified G3BP1 puncta. (**F**) Schematic summary of the findings in Figs. 2 and EV2. NT, untreated cells. Data, means ± SEM ($n = 3$); HCM: $n ≥ 3$ (each experiment: 500 valid primary objects/cells per well, ≥5 wells/sample). †$p ≥ 0.05$ (not significant), **$p < 0.01$, ANOVA. See also Fig. EV2. Source data are available online for this figure.

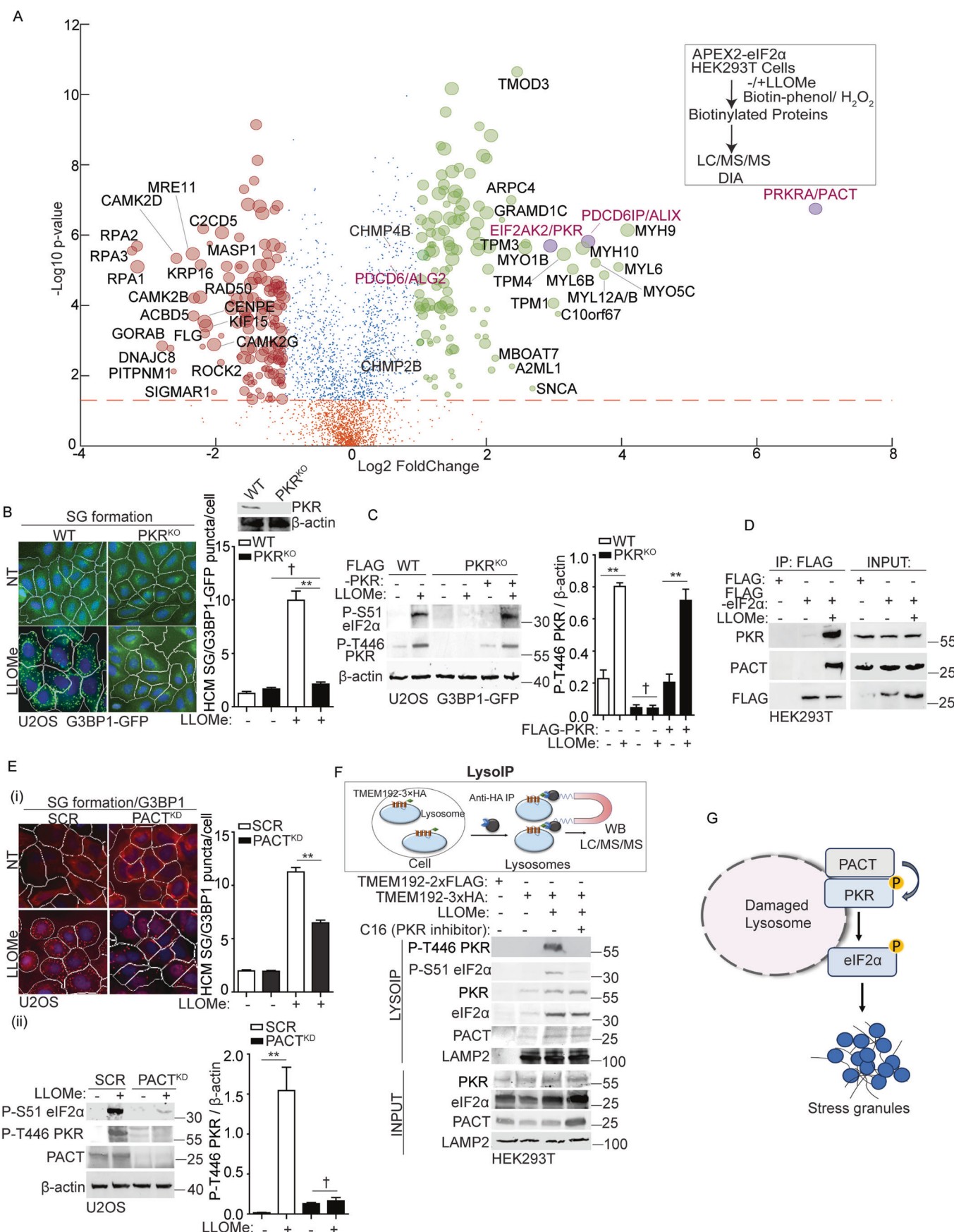

**Figure 3. PKR and its activator PACT regulate eIF2α phosphorylation on damaged lysosomes.**

(A) Quantitative liquid chromatography-tandem mass spectrometry (LC/MS/MS) using the data-independent acquisition (DIA) technique to identify eIF2α binding partners that were proximity-biotinylated by APEX2-eIF2α during lysosomal damage (1 mM LLOMe for 1 h). Scatter (volcano) plot shows log2 fold change (LLOMe/CTR; spectral counts) and –log10 p value for the proteins identified and quantified in three independent experiments. Green dots indicate increase in proximity to eIF2α (log2 fold change ≥ 1), and red dots indicate decrease in proximity to eIF2α (log2 fold change ≤ −1) during LLOMe treatment. Orange dots indicate values below the statistical significance cut-off (P ≥ 0.05). Bubble size represents a normalized value for the total amount of spectral counts for the protein indicated. PACT, PKR and ALIX proteins are highlighted as purple circles (see Dataset EV1). (B) Quantification by HCM of G3BP1-GFP puncta in wild type (WT) or PKR knockout (PKR^KO) U2OS G3BP1-GFP cells. Cells were treated with 2 mM LLOMe for 30 min. White masks, algorithm-defined cell boundaries; green masks, computer-identified G3BP1 puncta. (C) Immunoblot analysis of phosphorylation of eIF2α (S51) and PKR (T446) in WT or PKR^KO U2OS G3BP1-GFP cells, as well as in cells overexpressing FLAG-PKR in PKR^KO U2OS G3BP1-GFP cells. Cells were treated with 2 mM LLOMe for 30 min. The level of phosphorylation of PKR (T446) was quantified based on three independent experiments. (D) Co-IP analysis of interactions between eIF2α and PKR/PACT during lysosomal damage. HEK293T cells expressing FLAG (control) or FLAG-eIF2α were treated with 1 mM LLOMe for 30 min. Cell lysates were immunoprecipitated with anti-FLAG antibody and immunoblotted for indicated proteins. (E) (i) Quantification by HCM of G3BP1 puncta in U2OS cells transfected with either scrambled siRNA as control (SCR) or PACT siRNA for knockdown (PACT^KD). Cells were treated with 2 mM LLOMe for 30 min. White masks, algorithm-defined cell boundaries; red masks, computer-identified G3BP1 puncta; (ii) Immunoblot analysis of phosphorylation of eIF2α (S51) and PKR (T446) in SCR or PACT^KD cells; 2 mM LLOMe for 30 min. The level of phosphorylation of PKR (T446) was quantified based on three independent experiments. (F) Analysis of proteins associated with purified lysosomes (LysoIP; TMEM192-3xHA) from HEK293T cells treated with 1 mM LLOMe in the presence or absence of 210 nM imidazolo-oxindole C16 for 1 h. TMEM192-2xFLAG, control. The level of PKR, eIF2α and PACT in LysoIP was quantified based on three independent experiments shown in Fig. EV3B. (G) Schematic summary of the findings in Figs. 3 and EV3. NT, untreated cells. Data, means ± SEM (n = 3); HCM: n ≥ 3 (each experiment: 500 valid primary objects/cells per well, ≥5 wells/sample). †p ≥ 0.05 (not significant), **p < 0.01, ANOVA. See also Fig. EV3. Source data are available online for this figure.

expressing APEX2-eIF2α. We found that a 1 mM LLOMe treatment for 1 h initiated eIF2α phosphorylation in these cells without triggering cell death (Fig. EV2D). In contrast, treatment with 2 mM LLOMe for 30 min in U2OS cells initiated both eIF2α phosphorylation (Fig. EV2E) and the onset of cell death (Fig. 1A,B). Next, we identified and compared the interacting partners of eIF2α through LC/MS/MS in HEK293T cells expressing APEX2-eIF2α, under both control and 1 mM LLOMe 1 h treatment conditions (for a total of three independent experiments) to capture the early events of eIF2α phosphorylation (Dataset EV1). The volcano plot of this proteomic analysis showed dynamic changes in the proximity of cellular proteins to APEX2-eIF2α during lysosomal damage (Fig. 3A). Within the top twenty candidates showing increased association with eIF2α in response to lysosomal damage, we found the expected candidate PKR (EIF2AK2), which was previously reported by our group as a potential upstream kinase responsible for eIF2α phosphorylation during lysosomal damage (Jia et al, 2022) (Fig. 3A). Previously we knocked down four widely recognized upstream kinases of eIF2α (HRI, PKR, PERK, and GCN2) (Pakos-Zebrucka et al, 2016), and found that only the knockdown of PKR resulted in the inhibition of eIF2α phosphorylation and SG formation (Jia et al, 2022). Recently, MARK2 was identified as the fifth kinase responsible for eIF2α phosphorylation in response to proteotoxic stress (Lu et al, 2021). However, we found that MARK2 did not regulate eIF2α phosphorylation during lysosomal damage (Fig. EV2F). To confirm these findings, we generated a CRISPR knockout of PKR (PKR^KO) in SG reporter cells (U2OS G3BP1-GFP). In these PKR^KO cells, the formation of SG induced by lysosomal damage was completely inhibited, as quantified by the puncta of G3BP1-GFP using HCM (Fig. 3B). In line with this, the phosphorylation of eIF2α and PKR was also abolished (Fig. 3C). Conversely, the overexpression of PKR in PKR^KO cells led to a restoration of phosphorylation of eIF2α and PKR during lysosomal damage (Fig. 3C). It is known that PKR can be activated by double-stranded (ds) RNA or protein activator such as PACT (PRKRA) (Gal-Ben-Ari et al, 2019; Patel and Sen, 1998; Peters et al, 2001). First, we tested if dsRNA can regulate PKR activation during lysosomal damage. However, we could not detect the presence of dsRNA in response to LLOMe treatment (Fig. EV2G). In addition,

knocking down lysosomal RNase RNASET2 (Haud et al, 2011) did not affect the activation of PKR upon LLOMe treatment (Fig. EV2H), which aligns with our previous observation that RNASET2 did not affect SG formation during lysosomal damage (Jia et al, 2022). Furthermore, we generated a PKR mutant deficient in dsRNA-binding ability, PKR^K60A&K150A (McMillan et al, 1995; Patel et al, 1996). However, the overexpression of PKR^K60A&K150A in PKR^KO cells still led to the restoration of eIF2α phosphorylation and PKR activation during lysosomal damage (Fig. EV2I). Thus, these data suggest that dsRNA is not the trigger for PKR activation in response to lysosomal damage.

Interestingly, the protein activator of PKR, PACT, prominently emerged with the most significant fold increase following lysosomal damage (Fig. 3A). PACT is known to facilitate the stress-induced phosphorylation and activation of PKR through direct interaction (Patel and Sen, 1998; Singh and Patel, 2012). This interaction disrupts PKR's self-inhibition, leading to PKR autophosphorylation including at Thr446, which converts it into its fully active form capable of phosphorylating protein substrates, such as eIF2α (Chukwurah et al, 2021; Sadler and Williams, 2007). We confirmed increased interactions of PKR and PACT with eIF2α upon lysosomal damage by co-immunoprecipitation (co-IP) of FLAG-eIF2α with endogenous PKR and PACT (Fig. 3D). Next, we examined whether PKR and PACT are functionally necessary for eIF2α phosphorylation triggered by lysosomal damage. We observed a decrease in PKR activation, eIF2α phosphorylation, and SG formation observed in U2OS cells with PACT knockdown (PACT^KD) during lysosomal damage (Fig. 3E). Overexpression of PACT in PACT^KD cells restores PKR activation during lysosomal damage (Fig. EV2J). This finding aligns with the role of PKR in controlling eIF2α phosphorylation and SG formation. Thus, both PKR and its activator PACT regulate eIF2α phosphorylation for SG formation during lysosomal damage.

## PKR and PACT control eIF2α phosphorylation on damaged lysosomes

We previously performed proteomic analyses of lysosomes that were purified using LysoIP (Jia et al, 2022), a well-established approach to isolate lysosomes by the lysosomal membrane protein TMEM192

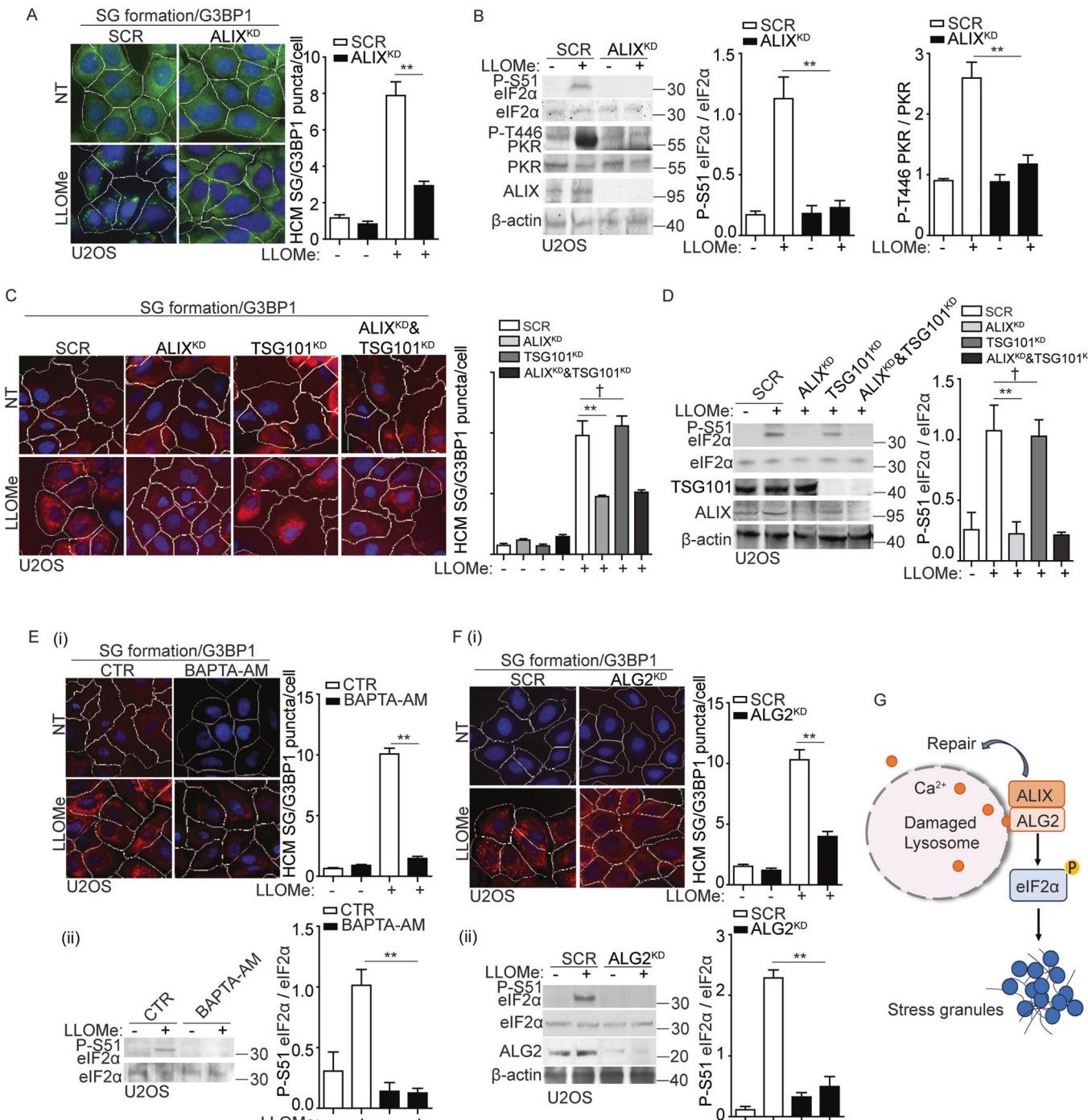

(Abu-Remaileh et al, 2017; Jia et al, 2020c). These analyses indicate the presence of PKR, PACT, and eIF2α on lysosomes (Fig. EV3A). This finding is further supported by similar results from LysoIP proteomic analysis conducted by other research groups (Eapen et al, 2021; Wyant et al, 2018) (Fig. EV3A). Using LysoIP immunoblotting, we confirmed the presence of PKR, PACT and eIF2α on lysosomes and found an elevation in their association with damaged lysosomes (Figs. 3F and EV3B). We also observed that the phosphorylation of both PKR and eIF2α occurred on damaged lysosomes (Fig. 3F). Notably, this effect

was effectively blocked by a specific PKR's inhibitor, imidazolo-oxindole C16, known for its ability to inhibit PKR's autophosphorylation by binding to PKR's ATP-binding pocket (Gal-Ben-Ari et al, 2019; Jammi et al, 2003; Tronel et al, 2014) (Fig. 3F). Moreover, through confocal fluorescence microscopy, an increased association of PKR, PACT, and eIF2α was detected with damaged lysosomes (Fig. EV3C–E). In summary, we conclude that PKR and its activator, PACT, regulate eIF2α phosphorylation on damaged lysosomes (Fig. 3G).

**Figure 4.   ALIX and ALG2 are required for stress granule formation by sensing calcium release from damaged lysosomes.**

(A) Quantification by HCM of G3BP1 puncta in U2OS cells transfected with either scrambled siRNA as control (SCR) or ALIX siRNA for knockdown (ALIX$^{KD}$). Cells were treated with 2 mM LLOMe for 30 min. White masks, algorithm-defined cell boundaries; green masks, computer-identified G3BP1 puncta. (B) Immunoblot analysis of phosphorylation of eIF2α (S51) and PKR (T446) in U2OS cells transfected with either scrambled siRNA as control (SCR) or ALIX siRNA for knockdown (ALIX$^{KD}$). Cells were treated with 2 mM LLOMe for 30 min. The level of phosphorylation of eIF2α (S51) and PKR (T446) was quantified based on three independent experiments. (C) Quantification by HCM of G3BP1 puncta in U2OS cells transfected with scrambled siRNA as control (SCR), ALIX siRNA for knockdown (ALIX$^{KD}$) or TSG101 siRNA for knockdown (TSG101$^{KD}$). Cells were treated with 2 mM LLOMe for 30 min. White masks, algorithm-defined cell boundaries; red masks, computer-identified G3BP1 puncta. (D) Immunoblot analysis of phosphorylation of eIF2α (S51) in U2OS cells transfected with scrambled siRNA as control (SCR), ALIX siRNA for knockdown (ALIX$^{KD}$) or TSG101 siRNA for knockdown (TSG101$^{KD}$). Cells were treated with 2 mM LLOMe for 30 min. The level of phosphorylation of eIF2α (S51) was quantified based on three independent experiments. (E) (i) Quantification by HCM of G3BP1 puncta in U2OS cells pre-treated with 15 μM BAPTA-AM for 1 h, subjected to 2 mM LLOMe treatment for 30 min. White masks, algorithm-defined cell boundaries; red masks, computer-identified G3BP1 puncta. (ii) Immunoblot analysis of phosphorylation of eIF2α (S51) in U2OS cells as described in (i) and was quantified based on three independent experiments. (F) (i) Quantification by HCM of G3BP1 puncta in U2OS cells transfected with scrambled siRNA as control (SCR), or ALG2 siRNA for knockdown (ALG2$^{KD}$). Cells were treated with 2 mM LLOMe for 30 min. White masks, algorithm-defined cell boundaries; red masks, computer-identified G3BP1 puncta. (ii) Immunoblot analysis of phosphorylation of eIF2α (S51) in U2OS cells as described in (i) and was quantified based on three independent experiments. (G) Schematic summary of the findings in Figs. 4 and EV4. NT, untreated cells. CTR, control. Data, means ± SEM ($n = 3$); HCM: $n \geq 3$ (each experiment: 500 valid primary objects/cells per well, ≥5 wells/sample). †$p \geq 0.05$ (not significant), **$p < 0.01$, ANOVA. See also Fig. EV4. Source data are available online for this figure.

## ALIX and ALG2 are required for stress granule formation by sensing calcium release from damaged lysosomes

In our proteomic analysis of eIF2α binding partners (Fig. 3A), we observed an increased association between eIF2α and ESCRT components such as ALIX, CHMP2B, and CHMP4B following lysosomal damage. Specifically, ALIX showed a greater than 10-fold increase (Fig. 3A). We next determined whether these ESCRT components were involved in eIF2α phosphorylation and SG formation triggered by lysosomal damage. Upon lysosomal damage, we observed a significant reduction in SG formation upon knockdown of ALIX in U2OS cells (ALIX$^{KD}$), as quantified by G3BP1 puncta using HCM (Fig. 4A,C). This was also reflected in the decreased phosphorylation of eIF2α and PKR in ALIX$^{KD}$ cells during LLOMe treatment (Fig. 4B,D), indicating an impact of ALIX on the upstream signaling of SG formation. In addition, we have tested the effect of ALIX knockdown on lysosomal biogenesis and observed no significant change in the overall number of lysosomes, as measured by the lysosomal marker LAMP2 in ALIX$^{KD}$ cells (Fig. EV4A). This aligns with our previous observations that the depletion of ALIX does not affect lysosomal function, as measured by the acidification of lysosomes using the Lysotracker assay and the activity of cathepsin B using the Magic Red assay (Jia et al, 2020c). However, the knockdown of CHM2B or CHMP4B had no discernible effect on SG formation and its upstream events (Fig. EV4B,C). Previous studies showed that the depletion of both ALIX and TSG101 effectively impedes lysosomal repair by eliminating ESCRT recruitment (Niekamp et al, 2022; Radulovic et al, 2018; Skowyra et al, 2018). We found that TSG101 has no effect on the regulation of SG formation upon lysosomal damage. This is supported by the absence of any significant changes in SG formation and eIF2α phosphorylation in TSG101 knockdown U2OS cells (TSG101$^{KD}$) (Fig. 4C,D). ALIX has been reported to sense lysosomal damage through the detection of calcium leakage, which is facilitated by its calcium binding partner, ALG2 (Chen et al, 2024; Jia et al, 2020a; Niekamp et al, 2022; Skowyra et al, 2018). Notably, ALG2 exhibited increased proximity to eIF2α upon lysosomal damage (Fig. 3A). To further determine the regulatory role of ALIX in SG formation upon lysosomal damage, we utilized BAPTA-AM, the calcium chelator and ALG2 knockdown U2OS cells (ALG2$^{KD}$) to prevent the recruitment of ALIX to damaged

lysosomes as previously reported (Jia et al, 2020a; Skowyra et al, 2018). This was confirmed by the observed decrease in ALIX puncta formation upon lysosomal damage in cells treated with BAPTA-AM or in ALG2$^{KD}$ cells (Fig. EV4D). Importantly, we also observed a significant reduction in SG formation and eIF2α phosphorylation in cells treated with BAPTA-AM, or in ALG2$^{KD}$ cells during lysosomal damage (Fig. 4E,F). Furthermore, overexpression of ALIX in ALIX$^{KD}$ cells and overexpression of ALG2 in ALG2$^{KD}$ cells both restored SG formation and eIF2α phosphorylation, respectively, during lysosomal damage (Fig. EV4E,F). Thus, we conclude that ALIX and its partner, ALG2, modulate eIF2α phosphorylation by sensing calcium leakage as lysosomal damage signal, thereby initiating SG formation (Fig. 4G).

## ALIX associates with PKR and PACT in response to lysosomal damage

Given that eIF2α phosphorylation is initiated by its upstream kinase PKR, and its activator PACT (Fig. 3), our subsequent investigation delved into exploring the relationship among ALIX, PKR and PACT. Using a co-IP assay, we tested the interaction between FLAG-ALIX and endogenous PKR and PACT. Their interactions were notably enhanced following treatment with LLOMe (Fig. 5A). ALIX is composed of three distinct domains: Bro1 domain, V domain, and proline-rich domain (PRD) (Fig. 5B). These domains have the potential to remain inactive due to intramolecular interactions but can be activated through interaction with ALG2 in a calcium-dependent manner (Maki et al, 2016; Scheffer et al, 2014; Sun et al, 2015; Vietri et al, 2020) (Fig. 5B). Next, we generated the domain deletions of ALIX (Fig. 5B(i)). The mapping analysis of ALIX domains necessary for binding to PKR and PACT revealed the indispensable role of the V domain in their interaction (Fig. 5C). In addition, increased associations among full-length ALIX, PKR and PACT were observed upon LLOMe treatment (Fig. 5A,C), suggesting that lysosomal damage activates ALIX by releasing its V domain for association with PKR and PACT. This is corroborated by the interaction of the V domain of ALIX with PKR and PACT, even in cells that were not subjected to lysosome damage induced by LLOMe (Fig. 5C). The interaction between the V domain of ALIX with PKR or PACT was also predicted using AlphaFold 2 (Jumper et al, 2021) (Fig. EV5A,B).

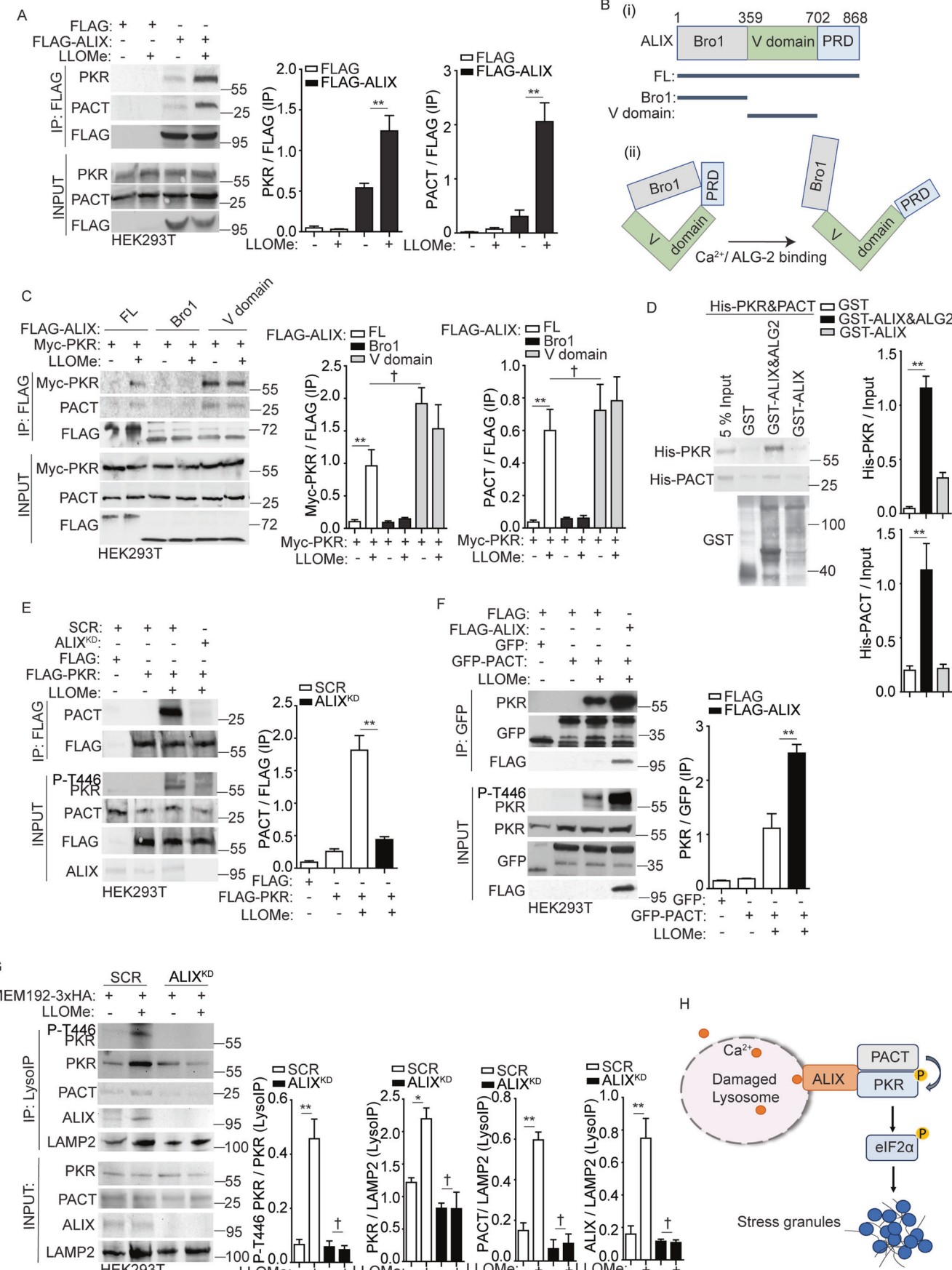

**Figure 5. ALIX promotes the association between PKR and its activator PACT on damaged lysosomes.**

(A) Co-IP analysis of interactions among ALIX, PKR and PACT during lysosomal damage. HEK293T cells expressing FLAG (control) or FLAG-ALIX were treated with 1 mM LLOMe for 30 min. Cell lysates were immunoprecipitated with anti-FLAG antibody and immunoblotted for indicated proteins. Quantification of IP analysis based on three independent experiments. (B) (i) Schematic diagram of ALIX mutants used in this study. FL (full length); Bro1 (Bro1 domain); V domain; PRD (proline-rich domain). Numbers, residue positions. (ii) Schematic illustration of the $Ca^{2+}$/ALG-2-induced open conformation of ALIX. (C) Co-IP analysis of interactions among ALIX mutants, PKR and PACT during lysosomal damage. HEK293T cells expressing FLAG tagged ALIX mutants and Myc-PKR were treated with 1 mM LLOMe for 30 min. Cell lysates were immunoprecipitated with anti-FLAG antibody and immunoblotted for indicated proteins. Quantification of IP analysis based on three independent experiments. (D) GST pulldown assay of in vitro translated His-tagged PKR and His-tagged PACT with GST, GST-tagged ALIX, with or without GST-tagged ALG2 in the presence of 10 μM $CaCl_2$. Quantification of the GST pulldown (the corresponding protein relative to its input) was performed based on three independent experiments. (E) Co-IP analysis of interactions between FLAG-PKR and PACT in HEK293T cells transfected with scrambled siRNA as control (SCR), or ALIX siRNA for knockdown (ALIX[KD]) during lysosomal damage. Cells were treated with 1 mM LLOMe for 30 min. Cell lysates were immunoprecipitated with anti-FLAG antibody and immunoblotted for indicated proteins. Quantification of IP analysis based on three independent experiments. (F) Co-IP analysis of interactions between PKR and GFP-PACT in HEK293T cells transfected with FLAG, or FLAG-ALIX during lysosomal damage. Cells were treated with 1 mM LLOMe for 30 min. Cell lysates were immunoprecipitated with anti-GFP antibody and immunoblotted for indicated proteins. Quantification of IP analysis based on three independent experiments. (G) Analysis of proteins associated with purified lysosomes (LysoIP; TMEM192-3xHA) from HEK293T cells transfected with scrambled siRNA as control (SCR), or ALIX siRNA for knockdown (ALIX[KD]). Cells were treated with 1 mM LLOMe for 30 min. Quantification of LysoIP analysis based on three independent experiments. (H) Schematic summary of the findings in Figs. 5 and EV5. See also Fig. EV5. †$p \geq 0.05$ (not significant), *$p < 0.05$, **$p < 0.01$, ANOVA. Source data are available online for this figure.

While co-IP results indicate that ALIX, PACT and PKR can form protein complexes (Fig. 5A,C), GST pulldown assays showed that ALIX or its partner ALG2 individually did not directly interact with PACT or PKR (Fig. EV5C,D). However, we found that ALIX and ALG2 together can directly interact with the PACT and PKR complex (Fig. 5D). This suggests that conformational changes, possibly induced by ALG2 exposing the V domain of ALIX (Sun et al, 2015) and PACT promoting PKR dimerization (Li et al, 2006), are important for the direct interaction. Nevertheless, a model emerges where ALIX and ALG2 interaction enables their direct binding to the PACT-PKR complex during calcium efflux caused by lysosome damage. Furthermore, by confocal fluorescence microscopy, we observed the association among ALIX, PKR, and PACT during lysosomal damage (Fig. EV5E). Thus, ALIX interacts with PKR and PACT in response to lysosomal damage.

## ALIX promotes the association between PKR and its activator PACT on damaged lysosomes

Next, we quantified by HCM the ALIX puncta response to lysosomal damage in cells where PKR or PACT had been knocked down. We observed that the presence or absence of PKR and PACT did not affect ALIX response to lysosomal damage (Fig. EV5F). This suggests that ALIX may potentially precede PKR and PACT for eIF2α phosphorylation upon lysosomal damage. Considering the decrease in the phosphorylation of PKR in ALIX[KD] cells and the increased association among ALIX, PKR, and PACT following lysosomal damage (Figs. 4B,D and 5A), we hypothesize that ALIX regulates PKR phosphorylation by modulating the association between PKR and its activator, PACT, during lysosomal damage. Using co-IP assays, we confirmed the formation of complexes between FLAG-PKR and endogenous PACT during lysosomal damage (Fig. 5E). However, this interaction was reduced in ALIX[KD] HEK293T cells (Fig. 5E), resulting in decreased PKR phosphorylation during LLOMe treatment. Conversely, the overexpression of ALIX led to a further enhancement in the increased association between GFP-PACT and endogenous PKR, and this was accompanied by an increase in PKR phosphorylation during lysosomal damage (Fig. 5F). These data indicates that ALIX is essential for PKR phosphorylation by controlling the interaction between PKR and PACT during lysosomal damage. Next, we examined whether

this regulatory event occurred on damaged lysosomes by conducting LysoIP immunoblotting in ALIX[KD] HEK293T cells. In this assay, we observed that ALIX[KD] HEK293T cells no longer displayed PKR phosphorylation on damaged lysosomes, accompanied by a reduced recruitment of PKR and PACT to lysosomes, as determined by Western blot analysis of lysosomes isolated using LysoIP (Fig. 5G). In contrast, ALIX overexpression in ALIX[KD] HEK293T cells reinstated PKR and PACT recruitment to damaged lysosomes and restored PKR phosphorylation on these organelles (Fig. EV5G). This suggests that ALIX is responsible for the recruitment and regulation of PKR and PACT on damaged lysosomes. In summary, we conclude that ALIX recruits PKR and its activator, PACT, to damaged lysosomes and regulates the activation of PKR by enhancing its association with PACT, consequently leading to eIF2α phosphorylation and SG formation (Fig. 5H).

## Galectin-3 inhibits stress granule formation by reducing the association between PKR and PACT during lysosomal damage

Previously, we reported that galectin-3 (Gal3), a β-galactoside-binding protein that recognizes damage-exposed glycan, can recruit ALIX to damaged lysosomes and promote ESCRT function for lysosomal repair and restoration (Jia et al, 2020d). We examined whether Gal3 is involved in the regulatory process of SG formation during lysosomal damage. In U2OS cells subjected to Gal3 knockdown (Gal3[KD]), we observed an elevated level of SG formation, quantified by the formation of G3BP1 puncta using HCM (Fig. 6A). This result was consistent with our earlier report showing an increase in SGs in Gal3 knockout HeLa cells (Jia et al, 2022). Here, we further detected the upstream signaling events leading to SG formation in Gal3[KD] U2OS cells and observed a significant increase in the phosphorylation of PKR and eIF2α in the absence of Gal3 following LLOMe treatment (Fig. 6B). These data indicate that Gal3 has a negative effect on the activation of PKR and eIF2α, thereby affecting SG formation during lysosomal damage. Next, the relationship among Gal3, PKR, and PACT was tested. The co-IP results showed that Gal3 can be in protein complexes with ALIX, PKR, and PACT upon lysosomal damage (Fig. 6C). When determining if Gal3 can control the association between PKR

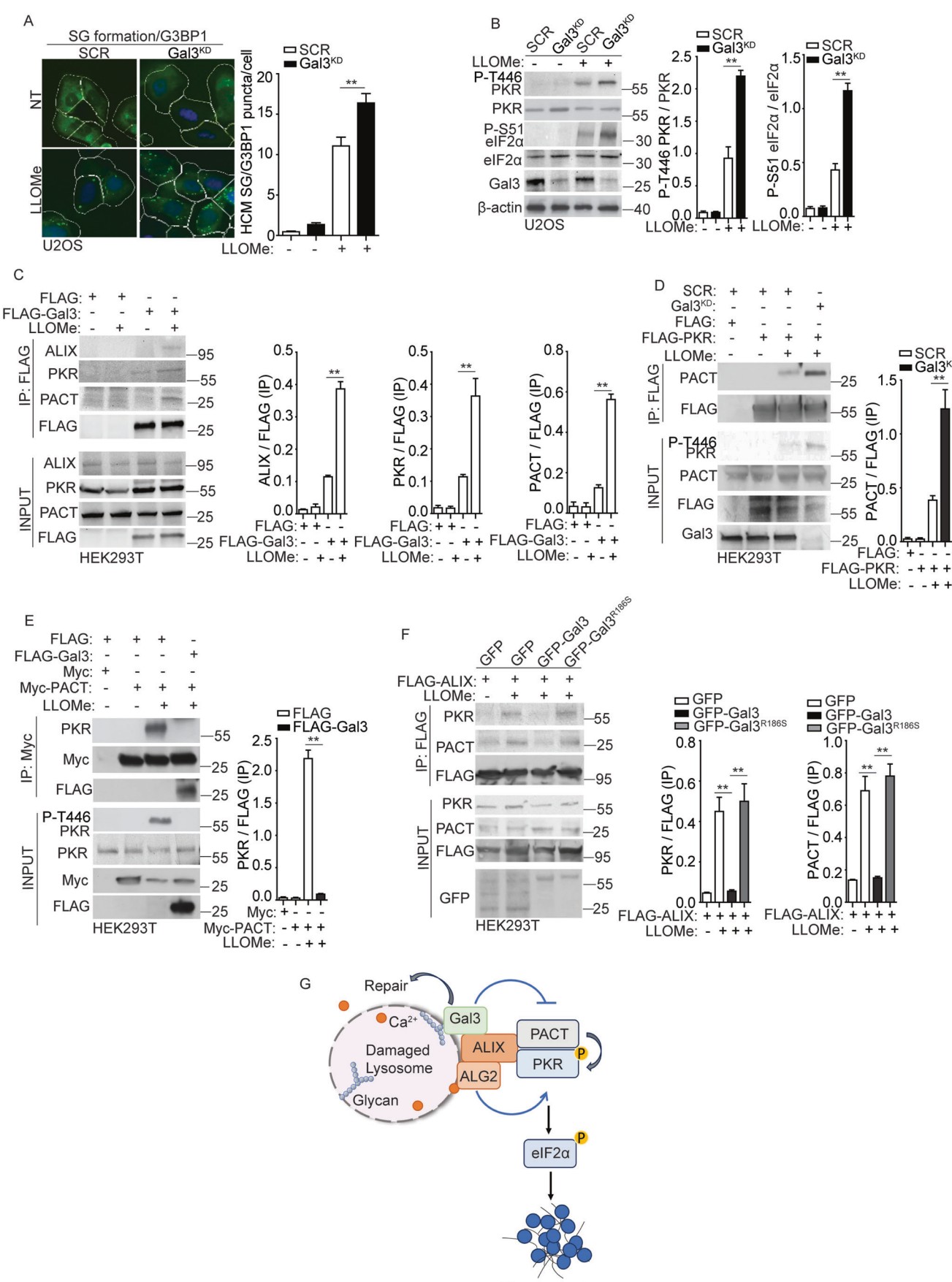

**Figure 6. Galectin-3 inhibits stress granule formation by reducing the association between PKR and PACT during lysosomal damage.**

(A) Quantification by HCM of G3BP1 puncta in U2OS cells transfected with scrambled siRNA as control (SCR), or galectin-3 (Gal3) siRNA for knockdown (Gal3$^{KD}$). Cells were treated with 2 mM LLOMe for 30 min. White masks, algorithm-defined cell boundaries; green masks, computer-identified G3BP1 puncta. (B) Immunoblot analysis of phosphorylation of eIF2α (S51) and PKR (T446) in U2OS cells transfected with scrambled siRNA as control (SCR), or galectin-3 (Gal3) siRNA for knockdown (Gal3$^{KD}$), subjected to 2 mM LLOMe treatment for 30 min. The level of phosphorylation of eIF2α (S51) and PKR (T446) was quantified based on three independent experiments. (C) Co-IP analysis of interactions among FLAG-Gal3, ALIX, PKR and PACT in HEK293T cells during lysosomal damage. Cells were treated with 1 mM LLOMe for 30 min. Cell lysates were immunoprecipitated with anti-FLAG antibody and immunoblotted for indicated proteins. Quantification of IP analysis for ALIX, PKR, and PACT based on three independent experiments. (D) Co-IP analysis of interactions between FLAG-PKR and PACT in HEK293T cells transfected with scrambled siRNA as control (SCR), or Gal3 siRNA for knockdown (Gal3$^{KD}$) during lysosomal damage. Cells were treated with 1 mM LLOMe for 30 min. Cell lysates were immunoprecipitated with anti-FLAG antibody and immunoblotted for indicated proteins. Quantification of IP analysis based on three independent experiments. (E) Co-IP analysis of interactions between Myc-PACT and PKR in HEK293T cells transfected with FLAG, or FLAG-Gal3 during lysosomal damage. Cells were treated with 1 mM LLOMe for 30 min. Cell lysates were immunoprecipitated with anti-Myc antibody and immunoblotted for indicated proteins. Quantification of IP analysis based on three independent experiments. (F) Co-IP analysis of interactions among FLAG-ALIX, PKR and PACT in HEK293T cells transfected with GFP, GFP-Gal3 or GFP-Gal3$^{R186S}$ during lysosomal damage. Cells were treated with 1 mM LLOMe for 30 min. Cell lysates were immunoprecipitated with anti-FLAG antibody and immunoblotted for indicated proteins. Quantification of IP analysis based on three independent experiments. (G) Schematic summary of the findings in Fig. 6. NT, untreated cells. Data, means ± SEM ($n = 3$); HCM: $n \geq 3$ (each experiment: 500 valid primary objects/cells per well, ≥5 wells/sample). **$p < 0.01$, ANOVA. Source data are available online for this figure.

and PACT, we found a marked increase in their association in the absence of Gal3 (Fig. 6D). This was further confirmed by the increased PKR phosphorylation under the same conditions. On the contrary, when Gal3 was overexpressed, it led to a significant reduction in the interaction between PKR and PACT, consequently resulting in reduced PKR phosphorylation upon LLOMe treatment (Fig. 6E). We interpret the inhibitory role of Gal3 in the association between PKR and PACT as a result of their competition for ALIX. Consistent with this interpretation, we observed a significantly reduced interaction among ALIX, PACT, and PKR in Gal3-overexpressing cells during LLOMe treatment (Fig. 6F). However, when we overexpressed the Gal3$^{R186S}$ mutant, which has been previously shown to lose the ability to recognize damaged lysosomes (Aits et al, 2015), it failed to regulate the protein complex of ALIX, PACT, and PKR upon lysosomal damage (Fig. 6F). Moreover, given our previous finding that Gal3 facilitates ESCRT-mediated lysosomal repair via ALIX (Jia et al, 2020d), these observations provide evidence of Gal3's role in balancing ALIX-mediated lysosomal repair and ALIX-mediated SG formation (Fig. 6G). Thus, we conclude that the recruitment of Gal3 to damaged lysosomes plays an inhibitory effect on the regulation of the upstream processes of SG formation by decreasing the association between PKR and PACT (Fig. 6G).

## Stress granule formation promotes cell survival in response to lysosomal damage in the context of disease states

Lysosomal damage serves as both a cause and consequence of many disease conditions, including infectious and neurodegenerative diseases (Amaral et al, 2023; Ballabio and Bonifacino, 2020; Bonam et al, 2019; Fehrenbacher et al, 2005). We tested whether the above molecular and cellular processes that transduce lysosomal damage signals to induce SG formation are important for cell survival in disease contexts. Lysosomal damage can occur from viral infections including those caused by non-enveloped adenovirus and enveloped SARS-CoV-2 infections (Aits et al, 2013; Barlan et al, 2011; Daussy and Wodrich, 2020; Thurston et al, 2012; Wang et al, 2018). Adenovirus enters cells through endocytosis and damages lysosomes by releasing its protease, which allows access to the cytosol and subsequently the nucleus for replication (Barlan et al, 2011; Greber et al, 1996; Pied et al, 2022; Wiethoff and Nemerow, 2015). We employed the wild type human adenovirus species C2 (HAdV-

C2$^{WT}$) and its protease-deficient mutant TS1 (HAdV-C2$^{TS1}$), the latter lacking the ability to damage lysosomes (Gallardo et al, 2021; Greber et al, 1996; Martinez et al, 2015). U2OS cells were infected with either HAdV-C2$^{WT}$ or HAdV-C2$^{TS1}$ and the lysosomal damage marker LysoTracker Red (LTR), which measures lysosomal acidification (Chazotte, 2011; Jia et al, 2020a; Pierzyńska-Mach et al, 2014), was quantified by HCM in infected cells. Consistent with earlier findings (Luisoni et al, 2015; Martinez et al, 2015; Pied et al, 2022), HAdV-C2$^{WT}$ led to a reduction in LTR$^+$ profiles, whereas HAdV-C2$^{TS1}$ did not show such an effect (Appendix Fig. S1A). In addition, SG formation and the phosphorylation of eIF2α and PKR were detected in cells infected with HAdV-C2$^{WT}$ but not in those infected with HAdV-C2$^{TS1}$ (Fig. 7A,B). These results imply that lysosomal damage triggered by HAdV-C2 infection can activate the PKR-eIF2α pathway, resulting in SG formation. We then tested whether SG formation is important for cell survival during HAdV-C2 infection. In SG-deficient U2OS (ΔΔG3BP1/2) cells, compared to wild-type U2OS cells, we observed an elevated level of cell death, using a PI uptake assay, during HAdV-C2$^{WT}$ infection (Fig. 7C). In addition, we expanded on our previous investigations showing that lysosomal damage induced by the expression of SARS-CoV-2 ORF3a protein (SARS-CoV-2$^{ORF3a}$) can also trigger SG formation (Jia et al, 2022). Following the overexpression of SARS-CoV-2$^{ORF3a}$ in U2OS cells, a notable rise in cell death was observed through an LDH release assay in ΔΔG3BP1/2 cells compared to control cells (Fig. 7D). Collectively, SG formation triggered by lysosomal damage emerges as a crucial process for cell survival during the viral infections examined.

In addition, other disease-associated agents in the context of human parasitic infections were examined that have the potential to damage lysosomes, such as malarial pigment (hemozoin). This parasitic agent is a crystalline and insoluble byproduct of hemoglobin digestion by *Plasmodial* species that is phagocytosed by circulating monocytes and neutrophils, and tissue macrophages, thus promoting immunopathological effects in human malaria (Anyona et al, 2022; Coronado et al, 2014; Guerra et al, 2019; Moore et al, 2004; Schwarzer et al, 1992; Weissbuch and Leiserowitz, 2008). Treatment of human monocytic THP-1 with physiological concentrations of hemozoin (0.1, 1.0, and 10.0 μg/mL) for 4 h, dose-dependently induced lysosomal damage, monitored by ALIX puncta formation serving as a lysosomal repair marker (Jia et al, 2020c; Radulovic et al, 2018; Skowyra et al, 2018) (Appendix Fig. S1B). While a previous report showed that hemozoin is rapidly ingested by human monocytes and exclusively

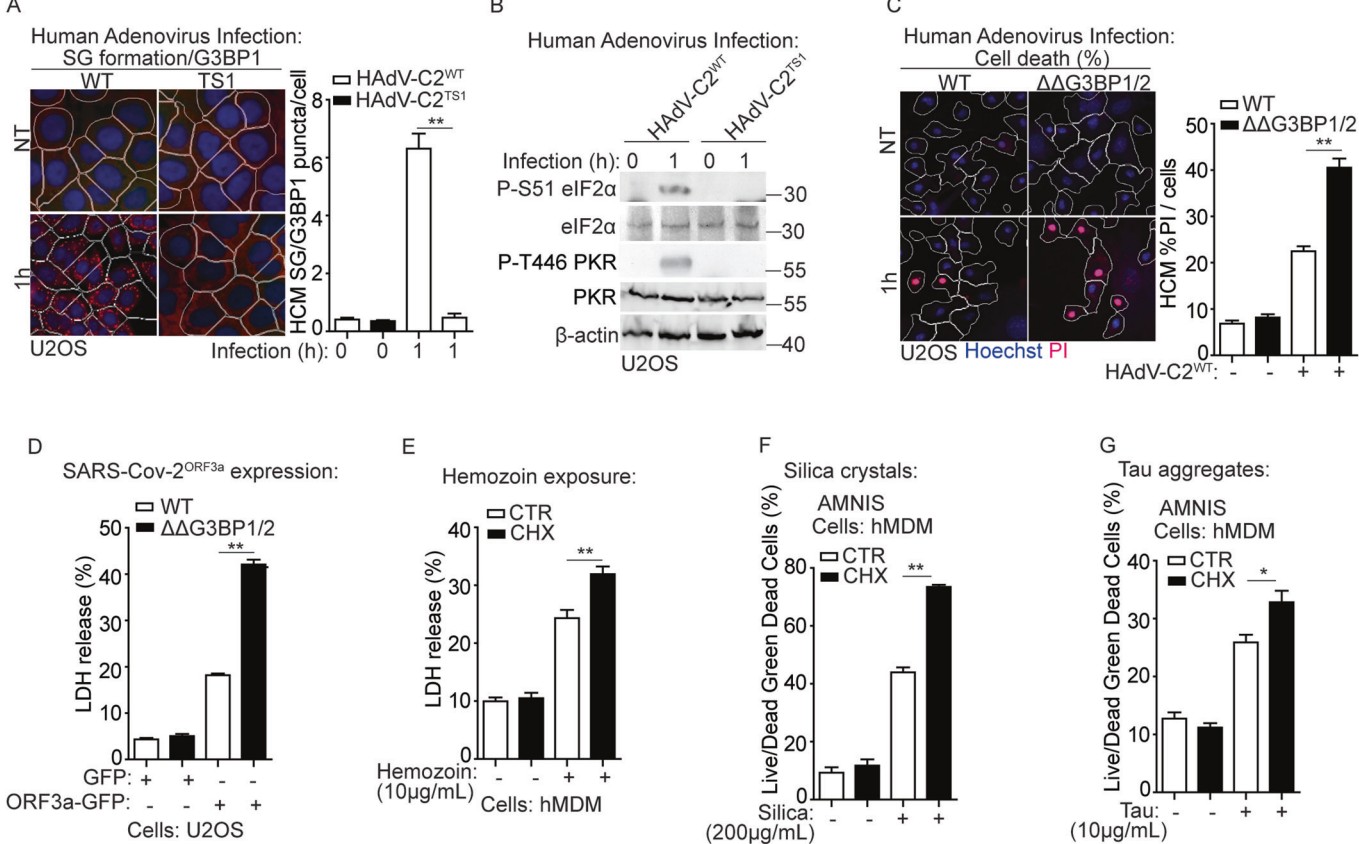

**Figure 7. Stress granule formation promotes cell survival in response to lysosomal damage during disease states.**

(A) Quantification by HCM of G3BP1 puncta in U2OS cells infected with wild-type human adenovirus C2 (HAdV-C2^WT) or C2 TS1 mutant (HAdV-C2^TS1) at MOI = 10 for 1 h. White masks, algorithm-defined cell boundaries; red masks, computer-identified G3BP1 puncta. (B) Immunoblot analysis of phosphorylation of eIF2α (S51) and PKR (T446) in U2OS cells infected with wild type human adenovirus C2 (HAdV-C2^WT) or C2 TS1 mutant (HAdV-C2^TS1) at MOI = 10 for 1 h. (C) Quantification by HCM of cell death by a propidium iodide (PI) uptake assay in U2OS wild type (WT) and G3BP1&2 double knockout (ΔΔG3BP1/2) cells during adenovirus infection. Cells were infected with wild-type human adenovirus C2 (HAdV-C2^WT) at MOI = 10 for 1 h, and then stained with propidium iodide PI (dead cells) and Hoechst-33342 (total cells). White masks, algorithm-defined cell boundaries; red masks, computer-identified PI⁺ nuclei. (D) Cell death analysis of supernatants of U2OS WT and ΔΔG3BP1/2 cells by a LDH release assay during SARS-Cov-2^ORF3a expression. Cells were transfected with the GFP-SARS-Cov-2^ORF3a construct overnight. (E) Cell death analysis of supernatants of human peripheral blood monocyte-derived macrophages (hMDM) by a LDH release assay during hemozoin exposure. Cells were treated with 10 μg/ml hemozoin for 4 h in the presence or absence of 1 μg/ml cycloheximide (CHX). (F) Quantification using AMNIS of cell death by Live/Dead™ stain kit in hMDM during silica treatment. Cells were treated with 200 μg/mL silica for 4 h in the presence or absence of 1 μg/ml cycloheximide (CHX), and then stained using Live/Dead™ stain kit (ThermoFisher). (G) Quantification using AMNIS of cell death by Live/Dead™ stain kit in hMDM during the treatment of tau oligomer. Cells were treated with 10 μg/mL tau oligomer for 4 h in the presence or absence of 1 μg/ml cycloheximide (CHX), and then stained using Live/Dead™ stain kit (ThermoFisher). CTR, control. Data, means ± SEM (n = 3); HCM: n ≥ 3 (each experiment: 500 valid primary objects/cells per well, ≥5 wells/sample). *p < 0.05, **p < 0.01, ANOVA. See also Appendix Fig. S1. Source data are available online for this figure.

localized in normally acidified phagolysosomes (Schwarzer et al, 2001), our findings suggest that hemozoin can perturb lysosomal membranes. Differences in the studies may be due to cell types, dosage, and treatment duration. In addition, stimulation with hemozoin (10.0 μg/mL) for 4 h, resulted in both SG formation and the phosphorylation of eIF2α and PKR (Appendix Fig. S1C, D). Blocking SG formation with cycloheximide in hMDM cells, showed an increased cell death as measured by LDH release assay in response to hemozoin treatment (10.0 μg/mL) (Fig. 7E). Moreover, we examined other lysosomal damaging agents, such as silica crystals associated with silicosis (Hornung et al, 2008; Mossman and Churg, 1998; Wang et al, 2017) and tau aggregates implicated in Alzheimer's disease (Flavin et al, 2017; Papadopoulos et al, 2017). We have previously reported that both silica crystals and tau aggregates induce lysosomal damage,

leading to SG formation (Jia et al, 2022). This effect was further confirmed by detecting the phosphorylation of eIF2α and PKR in hMDM cells in response to the treatment of silica crystals or tau aggregates (Appendix Fig. S1D). The prevention of SG formation with cycloheximide during the treatment of silica crystals or tau aggregates led to augmented cell death, as assessed using an AMNIS imaging flow cytometer in hMDM cells (Fig. 7F,G). Similarly, the application of another SG inhibitor, ISRIB to inhibit SG formation triggered by silica crystals or tau aggregate, produced a comparable effect on cell death of hMDM cells, measured by PI uptake assay (Appendix Fig. S1E,F). To further emphasize the role of the PKR-eIF2α pathway in controlling SG formation and promoting cell survival in disease contexts, we employed the PKR inhibitor C16 and assessed its impact on cell death in U2OS and hMDM cells via LDH release assay. We observed that

C16 treatment increased cell death when cells were exposed to various physiological agents that induce lysosomal damage (Appendix Fig. S1G–K). These findings also align with the role of SGs in promoting cell survival. In summary, our findings suggest that SG formation induced by lysosomal damage is important for cell survival against diverse pathogenic challenges associated with major human diseases.

## Discussion

In this study, we uncovered the regulation and significance of SG formation in response to lysosomal damage, providing insights into the interaction between membrane-bound organelles and membrane-less condensates. Through unbiased approaches, including proteomic analysis and high content microscopy, we defined a novel signaling pathway that transmits calcium leakage from damaged lysosomes to induce eIF2α phosphorylation, ultimately leading to SG formation, thus promoting cell survival. This study aligns with recent research indicating the role of SGs in plugging damaged membranes and aiding in lysosomal repair (Bussi et al, 2023), underscoring SG formation as a vital cellular protective mechanism against lysosomal damage, essential for survival.

How does the cell detect lysosomal damage to initiate SGs? Our study revealed the significant involvement of a calcium signal in this process. Lysosomes function as key intracellular calcium reservoirs for various cellular activities (Lloyd-Evans et al, 2020; Xu and Ren, 2015). We found that ALIX and ALG2 sense calcium leakage from damaged lysosomes, which activates ALIX's role in regulating PKR's activity. This ultimately leads to eIF2α phosphorylation and SG formation. In addition, our study indicates that PKR activation in response to lysosomal damage is independent of dsRNA but relies on its endogenous activator PACT. Under cellular stress, PACT directly binds to PKR, promoting PKR dimerization and conformational changes that lead to its autophosphorylation and enzymatic activation (Chukwurah et al, 2021). Notably, PKR activation is also observed in response to monosodium urate exposure (Lu et al, 2012), which is known to cause lysosomal damage (Maejima et al, 2013). Moreover, we found that ALIX controls the association between PKR and PACT, resulting in the phosphorylation of eIF2α. Importantly, we found that the role of ALIX and ALG2 in controlling eIF2α phosphorylation is distinct from their established function in ESCRT-mediated lysosomal repair. This suggests the multifaceted roles of ALIX and ALG2 as calcium sensors in coordinating cellular responses to lysosomal damage. Furthermore, our findings also indicate the intricate and adaptable nature of calcium signaling pathways in coordinating various cellular defense mechanisms against lysosomal damage. This extends beyond their involvement in TFEB nuclear translocation and phosphoinositide-mediated rapid lysosomal repair (Medina et al, 2015; Nakamura et al, 2020; Tan and Finkel, 2022).

SGs consist of RNA-binding proteins and untranslated mRNA, both playing a crucial role in the process of phase separation (Millar et al, 2023). In addition to the calcium signal we reported here as a trigger for SG formation during lysosomal damage, a recent study suggests that a decrease in pH can also induce SG formation on damaged lysosomes (Bussi et al, 2023). This is in line with the reported role of pH in G3BP1-driven SG condensation (Guillén-Boixet et al, 2020). However, the latter report indicates that pH may not directly regulate the RNA-binding affinity of G3BP1 but instead influences protein-protein interactions. It is worth noting that these experiments were conducted in an in vitro system and the presence of mRNA. Therefore, it raises the possibility that multiple mechanisms may collaborate to trigger SG formation by controlling protein-protein interaction or the accumulation of untranslated mRNA in response to lysosomal damage. To understand the signaling mechanism responsible for the accumulation of untranslated mRNA, our study suggests a calcium-dependent pathway that induces untranslated mRNA for SG formation by controlling eIF2α phosphorylation. Thus, both pH and calcium-dependent pathways can collaboratively contribute to SG formation during lysosomal damage. Moreover, considering the central role of lysosomes as the main degradation center for diverse cellular components (Lawrence and Zoncu, 2019), and the recognition of lysosomal damage that can be sensed by various cellular mechanisms (Aits et al, 2015; Jia et al, 2022; Napolitano and Ballabio, 2016; Chrisovalantis Papadopoulos et al, 2017), the leakage of certain lysosomal contents or the activation of other lysosomal damage sensors may also contribute to the activation of PKR, eIF2α phosphorylation, or the regulation of SG formation.

Phosphorylation of eIF2α is a key event in SG formation as it causes the shutdown in global translation and the accumulation of untranslated mRNA, which triggers the phase separation, ultimately leading to SG formation (Ivanov et al, 2019; Riggs et al, 2020). However, there are instances of SG formation that occur independently of eIF2α phosphorylation, potentially regulated by translational shutdown through the mTORC1 pathway (Emara et al, 2012; Fujimura et al, 2012). Nevertheless, this does not appear to be the case for SG formation in response to lysosomal damage. Our data indicate that upon lysosomal damage, eIF2α phosphorylation is the primary driver for SG formation, though the impact of mTORC1 inactivation on translation shutdown and SG formation cannot be entirely ruled out. Importantly, the uncoupled relationship between mTORC1 inactivation and eIF2α phosphorylation in SG formation may be attributed to their differential impacts on protein translation events and mRNA entry into SGs. For example, mTORC1 inactivation primarily inhibits the translation pre-initiation, while eIF2α phosphorylation can impede the recruitment of the large ribosomal subunit to mRNA (Holz et al, 2005; Jackson et al, 2010). Recent research suggests that having just one large ribosomal subunit on mRNA is enough to prevent the recruitment of mRNA into SGs, while extended ribosome-free regions on mRNA are insufficient for SG formation (Fedorovskiy et al, 2023). Thus, mTORC1 inactivation may result in ribosome-free regions on mRNA, but alone, it is insufficient to prompt mRNA entry into SGs. The prevention of large ribosomal subunits on mRNA through eIF2α phosphorylation appears to be a crucial factor triggering this process and contributing to SG formation in the context of lysosomal damage. In addition, through the examination of SG formation in galectin knockout cells, we recently showed (Jia et al, 2022) that galectin-8 does not influence SG formation. This finding supports the premise that eIF2α phosphorylation and mTORC1 inactivation are dissociated events during lysosomal damage, as we have previously reported that galectin-8 can modulate mTORC1 activity under similar conditions (Jia et al, 2018). Recent research has highlighted lysosomes as pivotal hubs in metabolic signaling, involving mTORC1 and AMPK pathways (Carroll and Dunlop, 2017; Jia et al, 2018; Jia et al, 2020b;

Zoncu et al, 2011). While PKR and eIF2α activation can occur in various cellular locations, our findings on eIF2α phosphorylation regulation on damaged lysosomes, combined with our earlier observations of mTORC1 inactivation on damaged lysosomes (Jia et al, 2018), suggest a novel role for lysosomes as central command centers in orchestrating protein translation signaling during stress conditions.

The understanding of how SGs contribute to cell survival during stress, especially in the context of lysosomal damage, remains limited. A recent report highlights the reparative role of SGs through their association with damaged lysosomes (Bussi et al, 2023). This finding aligns with our prior research; however, in our study, we observed SGs at a distance from damaged lysosomes (Jia et al, 2022). Our observation challenges the notion of SGs primarily serving as plugs and suggests a broader spectrum of roles for SGs in response to lysosomal damage. Given the significance of SG formation in supporting cell survival during lysosomal damage, as reported here, it is highly likely that SGs undertake multiple tasks in restoring cellular homeostasis for survival. For example, considering SGs sequester non-translating mRNA (Khong et al, 2017), they may play roles in protecting mRNA and controlling mRNA fate of the transcriptome during lysosomal damage. Moreover, SG formation intersects with the integrated stress response (ISR), which can optimize the cell response by reprogramming gene expression to promote cellular recovery (Pakos-Zebrucka et al, 2016). The impact of SG formation on ISR may also enhance cellular fitness. In addition, the involvement of SGs in various cellular processes, e.g., intracellular transport dynamics, ribosome biogenesis, and cell signaling (Gorsheneva et al, 2024; Ripin et al, 2023; Zhang et al, 2024), may further contribute to cell survival upon lysosomal damage.

Recognizing lysosomal damage as a critical internal physiological trigger for SGs highlights the importance of enhancing our understanding of SG formation in disease contexts. We detected the role of SG formation in cell survival within disease-specific contexts using a series of pathological reagents to induce lysosomal damage. Given the strong association of these reagents with both lysosomal damage and SG formation, delving into the molecular mechanisms governing the interaction between lysosomal damage and SGs may provide valuable insights for future therapeutic efforts.

## Methods

### Antibodies and reagents

Antibodies from Cell Signaling Technology were Phospho-eIF2α (Ser51) (1:1000 for WB), eIF2α (1:1000 for WB), Phospho-p70 S6 Kinase (Thr389)(108D2) (1:1000 for WB), p70 S6 Kinase (49D7) (1:1000 for WB), 4EBP1 (1:1000 for WB), Phospho-4EBP1 (Ser65) (1:1000 for WB), Phospho-ULK1 (Ser757) (1:1000 for WB), ULK1 (D8H5) (1:1000 for WB), TFEB (1:1000 for WB), GST(91G1) (1:1000 for WB), PKR (1:1000 for WB), PACT (D9N6J) (1:1000 for WB), Myc (9B11) (1:1000 for WB), mTOR (7C10) (1:1000 for WB; 1:400 for IF), ATF4 (D4B8) (1:1000 for WB) and G3BP2 (1:1000 for WB). Antibodies from Abcam were GFP (ab290) (for immunoprecipitation (IP) or 1:1000 for WB), CHMP4B (ab105767) (1:1000 for WB), TSG101 (4A10) (ab83) (1:1000 for WB), PKR (phospho T446) (E120) (ab32036) (1:1000 for WB). Antibodies from Sigma-Aldrich: FLAG M2 (F1804) (for IP and 1:1000 for WB), dsRNA J2 (MABE1134) (1:200 for IF), RNASET2 (HPA029013) (1:1000 for WB), phospho TFEB (Ser142; 1:1000 for WB). Antibodies from Proteintech: EIF4G1 (15704-1-AP) (1:200 for IF), CHMP2A (10477-1-AP) (1:500 for WB), MARK2 (15492-1-AP) (1:500 for WB) and ALG2 (15092-1-AP)

**Reagents and tools table**

| Reagent/Resource | Reference or Source | Identifier or Catalog Number |
|---|---|---|
| **Experimental Models cell lines** | | |
| U2OS WT and ΔΔG3BP1/2 | (Kedersha et al, 2016) | N/A |
| U2OS Flp-In | This study | N/A |
| hMDM (human peripheral blood monocyte-derived macrophages) | This study | N/A |
| U2OS G3BP1-GFP PKR^KO | This study | N/A |
| HEK293T | ATCC | CRL-3216 |
| HEK293T-TMEM192-2xFLAG | This study | N/A |
| HEK293T-TMEM192-3xHA | This study | N/A |
| THP-1 | THP-1 | TIB-202 |
| HEK293T-APEX2-eIF2α | This study | N/A |
| **Recombinant DNA** | | |
| pLJC5-TMEM192-3xHA | Addgene | #102930 |
| pLJC5-TMEM192-2xFLAG | Addgene | #102929 |
| pCMV-VSV-G | Addgene | #8454 |
| psPAX2 | Addgene | #12260 |
| pOG44 | This work | N/A |
| eIF2α1 | Addgene | #21807 |

| Reagent/Resource | Reference or Source | Identifier or Catalog Number |
|---|---|---|
| eIF2α2 | Addgene | #21808 |
| pDEST-FLAG-G3BP1 | This work | N/A |
| pDEST-FLAG-G3BP2 | This work | N/A |
| pDEST-GFP-G3BP1 | This work | N/A |
| pDEST-Flp-G3BP1-GFP | This work | N/A |
| pDEST-FLAG-RagB | (Jia et al, 2018) | N/A |
| pDEST- FLAG-RagB$^{Q99L}$ | (Jia et al, 2018) | N/A |
| pDEST-FLAG | This work | N/A |
| pDEST-GFP | This work | N/A |
| pDEST-GFP-eIF2α | This work | N/A |
| pDEST-FLAG-eIF2α | This work | N/A |
| pDEST-FLAG-eIF2A$^{S51A}$ | This work | N/A |
| pDEST-FLAG-PKR | This work | N/A |
| pDEST-Myc-PKR | This work | N/A |
| pDEST-GFP-PKR | This work | N/A |
| pDEST-GFP-PKR$^{K60A\&K150A}$ | This work | N/A |
| pDEST-FLAG-ALIX | This work | N/A |
| pDEST-FLAG-ALG2 | This work | N/A |
| pDEST-GFP-PACT | This work | N/A |
| pDEST-FLAG-PACT | This work | N/A |
| pDEST-FLAG-Gal3 | (Jia et al, 2020c) | N/A |
| pDEST-GFP-Gal3 | (Jia et al, 2020c) | N/A |
| pDEST-GFP-Gal3$^{R186S}$ | (Jia et al, 2020c) | N/A |
| pDEST-GFP-ORF3a | This work | N/A |
| **Antibodies** | | |
| Rabbit Phospho-eIF2α (Ser51) | Cell Signaling Technology | #9721 |
| Rabbit eIF2α | Cell Signaling Technology | #9722 |
| Rabbit Phospho-p70 S6 Kinase (Thr389) (108D2) | Cell Signaling Technology | #9234 |
| Rabbit p70 S6 Kinase (49D7) | Cell Signaling Technology | #2708 |
| Rabbit Phospho-4EBP1(Ser65) | Cell Signaling Technology | #9451 |
| Rabbit 4EBP1 | Cell Signaling Technology | #9644 |
| Rabbit TFEB | Cell Signaling Technology | #4240 |
| Rabbit Phospho-ULK1 (Ser757) | Cell Signaling Technology | #6888 |
| Rabbit ULK1 (D8H5) | Cell Signaling Technology | #8054 |
| Rabbit GST (91G1) | Cell Signaling Technology | #5475 |
| Rabbit PKR | Cell Signaling Technology | #3072 |
| Rabbit PACT (D9N6J) | Cell Signaling Technology | #13490 |
| Mouse Myc (9B11) | Cell Signaling Technology | #2276 |
| Rabbit mTOR (7C10) | Cell Signaling Technology | #2983 |
| Rabbit ATF4(D4B8) | Cell Signaling Technology | #11815 |
| Rabbit G3BP2 | Cell Signaling Technology | #31799 |
| Rabbit G3BP1 | Cell Signaling Technology | #17798 |
| Rabbit Anti-GFP (ab290) | Abcam | ab290 |
| Rabbit CHMP4B | Abcam | ab105767 |

| Reagent/Resource | Reference or Source | Identifier or Catalog Number |
|---|---|---|
| Rabbit TSG101(4A10) | Abcam | ab83 |
| Rabbit PKR (phospho T446) (E120) | Abcam | ab32036 |
| Mouse Anti-FLAG M2 | Sigma-Aldrich | F1804 |
| Mouse dsRNA J2 | Sigma-Aldrich | MABE1134 |
| Rabbit Anti-RNASET2 | Sigma-Aldrich | #HPA029013 |
| Rabbit phospho TFEB (Ser142) | Sigma-Aldrich | #ABE1971 |
| Rabbit EIF4G1 | Proteintech | 15704-1-AP |
| Rabbit CHMP2A | Proteintech | 10477-1-AP |
| Rabbit MARK2 | Proteintech | 15492-1-AP |
| Rabbit ALG2 | Proteintech | 15092-1-AP |
| Mouse Anti-Galectin-3 | BioLegend | #125402 |
| Mouse Anti-ALIX | BioLegend | #634502 |
| Mouse LAMP2 | DSHB of University of Iowa | H4B4 |
| Rabbit beta-Actin (C4) | Santa Cruz Biotechnology | sc-47778 |
| Mouse 6x-His Tag Monoclonal Antibody | ThermoFisher | MA1-21315 |
| Rabbit Anti-G3BP1 | ThermoFisher | #PA5-29455 |
| Alexa Fluor 488 Goat anti-Rabbit secondary antibody | ThermoFisher | #A-11034 |
| Alexa Fluor 488 Goat anti-Mouse secondary antibody | ThermoFisher | #A-11029 |
| Alexa Fluor 568 Goat anti-Rabbit secondary antibody | ThermoFisher | #A-11011 |
| Alexa Fluor 568 Goat anti-Mouse secondary antibody | ThermoFisher | #A-11004 |
| Alexa Fluor 647 Goat anti-Rabbit secondary antibody | ThermoFisher | #A27040 |
| Alexa Fluor 647 Goat anti-Rat secondary antibody | ThermoFisher | #A-21247 |
| Goat anti-rabbit IgG-HRP secondary antibody | ThermoFisher | #31460 |
| Goat anti-mouse IgG-HRP secondary antibody | ThermoFisher | #31430 |
| **Oligonucleotides and other sequence-based reagents** | | |
| PKR$^{K60A\&K150A}$ mutant oligonucleotide sense<br>5'-CGGCATTTTTTGCTTCCTTCGCTGATCTACCTTCACCTTCTG-3'<br>5'-AATTGTTTTGCTTCCTGTGCAGTAGAACCTGTACCAATACTATATTCTTTCTG-3' | GENEWIZ | N/A |
| PKR$^{K60A\&K150A}$ mutant oligonucleotide anti-sense<br>5'-CAGAAGGTGAAGGTAGATCAGCGAAGGAAGCAAAAAATGCCG-3'<br>5'-CAGAAAGAATATAGTATTGGTACAGGTTCTACTGCACAGGAAGCAAAACAATT-3' | GENEWIZ | N/A |
| eIF2α gateway oligonucleotide sense<br>5'-GGGGACAAGTTTGTACAAAAAAGCAGGCTTCATGCCGGGTCTAAGTTGTAGATTTTATC-3' | GENEWIZ | N/A |
| eIF2α gateway oligonucleotide anti-sense<br>5'-GGGGACCACTTTGTACAAGAAAGCTGGGTCTTAATCTTCAGCTTTGGCTTCCATTTC-3' | GENEWIZ | N/A |
| ALIX mutants oligonucleotide:<br>FL/Bro1 sense: 5'-GGGGACAAGTTTGTACAAAAAAGCAGGCTTCGCGACATTCATCTCGGTGCAGCTG-3'<br>FL anti-sense: 5'-GGGGACCACTTTGTACAAGAAAGCTGGGTCTTACTGCTGTGGATAGTAAGACTG-3'<br>Bro anti-sense: 5'-GGGGACCACTTTGTACAAGAAAGCTGGGTCTTAAACCATCTTCTCAAACAGATC-3'<br>V domain sense: 5'-GGGGACAAGTTTGTACAAAAAAGCAGGCTTCCCCGTGTCAGTACAGCAGTC-3'<br>V domain anti-sense: 5'-GGGGACCACTTTGTACAAGAAAGCTGGGTCTCTTTCTGTCTTCCGTGCAAAAAC-3' | GENEWIZ | N/A |
| siGENOME Non-Targeting Control siRNA | Horizon Discovery | D-001210-01-05 |
| siGENOME human G3BP1 SMARTpool siRNA | Horizon Discovery | L-012099-00-0005 |
| siGENOME human G3BP2 SMARTpool siRNA | Horizon Discovery | L-015329-01-0005 |
| siGENOME human EIF2S1(eIF2α) SMARTpool siRNA | Horizon Discovery | L-015389-01-0005 |

| Reagent/Resource | Reference or Source | Identifier or Catalog Number |
|---|---|---|
| siGENOME human EIF2AK2 (PKR) SMARTpool siRNA | Horizon Discovery | M-003527-00-0005 |
| siGENOME human PRKRA (PACT) SMARTpool siRNA | Horizon Discovery | L-006426-00-0005 |
| siGENOME human PDCD6IP (ALIX) SMARTpool siRNA | Horizon Discovery | L-004233-00-0005 |
| siGENOME human TSG101 SMARTpool siRNA | Horizon Discovery | L-003549-00-0005 |
| siGENOME human PDCD6 (ALG2) SMARTpool siRNA | Horizon Discovery | L-004440-00-0005 |
| siGENOME human CHMP2B SMARTpool siRNA | Horizon Discovery | L-004700-01-0005 |
| siGENOME human CHMP4B SMARTpool siRNA | Horizon Discovery | L-018075-01-0005 |
| siGENOME human LGALS3 (galectin-3) SMARTpool siRNA | Horizon Discovery | R-010606-00-0005 |
| siGENOME human RNASET2 SMARTpool siRNA | Horizon Discovery | M-009282-01-0005 |
| siGENOME human MARK2 SMARTpool siRNA | Horizon Discovery | L-004260-00-0005 |
| EIF2AK2 (PKR) CRISPR gRNA: gRNA1: GATGGAAGAGAATTTCCAGA gRNA2: AGTGTGCATCGGGGGTGCAT gRNA3: TGGTACAGGTTCTACTAAAC | Applied Biological Materials | 19075111 |
| **Chemicals, Enzymes and other reagents** | | |
| Leu-Leu-methyl ester hydrobromide (LLOMe) | Sigma-Aldrich | L7393 |
| Tetracycline hydrochloride | Sigma-Aldrich | T3383 |
| Puromycin dihydrochloride | Sigma-Aldrich | P9620 |
| Human Macrophage Colony-Stimulating Factor | Sigma-Aldrich | M6518 |
| Imidazolo-oxindole PKR inhibitor C16 | Sigma-Aldrich | I9785 |
| Puromycin dihydrochloride | Sigma-Aldrich | P8833 |
| cycloheximide | Sigma-Aldrich | C4859 |
| ISRIB | Sigma-Aldrich | SML0843 |
| Sodium (meta)arsenite | Sigma-Aldrich | S7400 |
| PMSF | Sigma-Aldrich | 93482 |
| Silica crystal | US Silica | MIN-U-SIL-15 |
| 5'-Cy3-Oligo d(T)30 | GeneLink | 26-4330-02 |
| Hoechst 33342 | ThermoFisher | H3570 |
| Prolong Gold Antifade Mountant with DAPI | ThermoFisher | P36931 |
| BAPTA-AM | ThermoFisher | B1205 |
| LIVE/DEAD™ Fixable Green Dead Cell Stain Kit | ThermoFisher | L34960 |
| GST Protein Interaction Pull-Down Kit | ThermoFisher | 21516 |
| Flp-In™ Complete System | ThermoFisher | K601001 |
| Anti-HA Magnetic Beads | ThermoFisher | 88836 |
| Dynabeads Protein G | ThermoFisher | 10003D |
| Streptavidin Magnetic Beads | ThermoFisher | 88816 |
| LysoTracker Red DND-99 | ThermoFisher | L7528 |
| Human M-CSF Recombinant Protein | ThermoFisher | 300-25 |
| Propidium Iodide (PI) solution | ThermoFisher | P3566 |

| Reagent/Resource | Reference or Source | Identifier or Catalog Number |
|---|---|---|
| LR Clonase Plus Enzyme Mix | ThermoFisher | 11791100 |
| BP Clonase Plus Enzyme Mix | ThermoFisher | 11789100 |
| Lipofectamine RNAiMAX Transfection Reagent | ThermoFisher | 13778030 |
| Lipofectamine 2000 Transfection Reagent | ThermoFisher | 12566014 |
| One Shot Mach1 Phage-Resistant Competent *E.coli* | ThermoFisher | C862003 |
| NP40 Cell Lysis Buffer | ThermoFisher | FNN0021 |
| ProFection Mammalian Transfection System | Promega | E1200 |
| CytoTox 96® Non-Radioactive Cytotoxicity Assay | Promega | G1780 |
| FuGENE HD Transfection Reagent | Promega | E2311 |
| NEB 5-alpha Competent *E.coli* (High Efficiency) | New England Biolabs | C2987 |
| Protease Inhibitor Cocktail Tablets | Roche | 11697498001 |
| Poly (I:C) | InvivoGen | tlrl-pic |
| FAZ3532 | MedChemExpress | HY-162288 |
| FAZ3780 | MedChemExpress | HY-162289 |
| Recombinant Human ALIX protein | Abcam | ab132534 |
| Recombinant Human ALG2 Protein | NOVUS | H00085365-P01 |
| Recombinant GST Epitope Tag Protein | NOVUS | NBC1-18537 |
| Recombinant Human PACT His Protein | NOVUS | NBP2-51787 |
| Recombinant Human PKR Protein | LSBio | LS-G22902-20 |
| **Software** | | |
| iDEV software | ThermoFisher | N/A |
| AIM software | Carl Zeiss | N/A |
| Spectronaut software | Biognosys Inc | N/A |
| MATLAB software | MathWorks | N/A |
| **Deposited Data** | | |
| Raw MS DIA data | https://massive.ucsd.edu | MSV000088152 |
| Raw MS DIA data | http://www.proteomexchange.org | PXD028745 |
| Source data | BioStudies https://www.ebi.ac.uk/biostudies/studies/S-BSST1652 | S-BSST1652 |

(1:500 for WB). Antibodies from BioLegend: Galectin-3 (1:1000 for WB; 1:500 for IF) and ALIX (1:200 for IF). G3BP1 (PA5-29455, 1:1000 for WB, 1:200 for IF), His (MA1-21315, 1:1000 for WB), Alexa Fluor 488, 568, 647 (1:500 for IF) and secondary antibodies from Thermo-Fisher Scientific. Other antibodies used in this study were from the following sources: beta-Actin (C4) (1:1000 for WB) from Santa Cruz Biotechnology; LAMP2 (H4B4) (1:500 for IF) from DSHB of University of Iowa.

Reagents from Sigma-Aldrich were Leu-Leu-methyl ester hydrobromide (LLOMe), Sodium(meta)arsenite, Puromycin dihydrochloride, Imidazolo-oxindole PKR inhibitor C16, ISRIB and cycloheximide. Reagents from ThermoFisher were Hoechst 33342, BAPTA-AM, LIVE/DEAD™ Fixable Green Dead Cell Stain Kit, Lipofectamine RNAiMAX Transfection Reagent, BP/LR Clonase Plus Enzyme Mix, Prolong Gold Antifade Mountant with DAPI, Human M-CSF Recombinant Protein, Propidium Iodide (PI) solution, DMEM, Opti-MEM Reduced Serum Media, EBSS, PBS, Penicillin-Streptomycin, Fetal Bovine Serum, NP40 Cell Lysis Buffer, Anti-HA Magnetic Beads, Dynabeads Protein G, Streptavidin Magnetic Beads, LysoTracker Red DND-99 and GST Protein Interaction Pull-Down Kit (21516). The Reagents from Promega were CytoTox 96® Non-Radioactive Cytotoxicity Assay, FuGENE HD Transfection Reagent and ProFection Mammalian Transfection System. Other reagents used in this study were from the following sources: Poly (I:C) from InvivoGen (tlrl-pic); 5′-Cy3-Oligo d(T)30 from GeneLink (26-4330-02); Silica crystal from US Silica (MIN-U-SIL-15); Protease Inhibitor from Roche (11697498001). FAZ3532 and FAZ3780 from MedChemExpress (HY-162288 and HY-162289). Recombinant Human ALIX protein (ab132534) from Abcam; Recombinant Human ALG2 Protein (H00085365-P01), Recombinant GST Epitope Tag Protein (NBC1-18537), Recombinant Human PACT His Protein (NBP2-51787) from NOVUS; Recombinant Human PKR Protein (LS-G22902-20) from LSBio. Wild-type human

adenovirus species C2 (HAdV-C2$^{WT}$) and its protease-deficient mutant TS1 (HAdV-C2$^{TS1}$) were provided by Dr. Jaya Rajaiya (University of New Mexico Health Sciences Center, Albuquerque, NM). Hemozoin was prepared according to reported methods (Keller et al, 2004). Tau aggregates were provided by Dr. Kiran Bhaskar (University of New Mexico Health Sciences Center, Albuquerque, NM).

## Cells and cell lines

U2OS, HEK293T and THP-1 cells were from ATCC. Human peripheral blood monocyte-derived macrophages (hMDM) were derived from peripheral blood mononuclear cells (PBMCs) isolated from venipuncture blood from anonymous donors, details below. Cell lines for LysoIP were generated using constructs obtained from Addgene, details below. Knockout cell lines were generated by CRISPR/Cas9-mediated knockout system, and knockdown cell lines were generated by small interfering RNAs (siRNAs) from GE Dharmacon (siGENOME SMART pool), details below. U2OS G3BP1-GFP cell line was generated using Flp-In system (Thermo-Fisher), details below. U2OS wild type (WT) and G3BP1&2 double knockout (ΔΔG3BP1/2) cells were from Dr. Pavel Ivanov (Brigham and Women's Hospital and Harvard Medical School, Boston, MA).

## Cultured human peripheral blood monocyte-derived macrophages

40–50 mL of venipuncture blood was collected from healthy, consenting adult volunteers at the Vitalant Blood Donation Center (Albuquerque, NM). Blood from individual donors (10 mL vacutainer tubes) was placed into two 50 mL conical tubes and the volume was brought to 50 mL with sterile 1 X PBS followed by mixing inversely. 25 mL of the blood mix were carefully layered onto 20 mL of Ficoll (Sigma, #1077) in separate conical tubes and centrifuged at 2000 rpm for 30 min at 22 °C. The buffer layer containing human peripheral blood monocytes (PBMCs) was removed, pooled, washed with 1X PBS twice, and resuspended in 20 mL RPMI media with 10% human AB serum and Primocin. PBMCs were cultured in RPMI 1640 with GlutaMAX and HEPES (Gibco), 20% FBS, and 200 ng/mL Human M-CSF Recombinant Protein (ThermoFisher). Six days after the initial isolation, differentiated macrophages were detached in 0.25% Trypsin-EDTA (Gibco) and seeded for experiments.

## Plasmids, siRNAs, and transfection

Plasmids used in this study, e.g., eIF2α, ALIX, PKR, and PACT cloned into pDONR221 using BP cloning, and expression vectors were made utilizing LR cloning (Gateway, ThermoFisher) in appropriate pDEST vectors for immunoprecipitation assay. PKR mutants were generated utilizing the QuikChange site-directed mutagenesis kit (Agilent) and confirmed by sequencing (Genewiz). The codon-optimized gene (VectorBuilder and Genewiz) was used to rescure the knockdown cells. Small interfering RNAs (siRNAs) were from Horizon Discovery (siGENOME SMART pool). Plasmid transfections were performed using the ProFection Mammalian Transfection System, FuGENE® HD Transfection Reagent (Promega), or Lipofectamine 2000 Transfection Reagent (Thermo-Fisher). siRNAs were delivered into cells using Lipofectamine RNAiMAX (ThermoFisher).

## Generation of CRISPR mutant cells

PKR knockout cells were generated by CRISPR/Cas9-mediated knockout system. The lentiviral vector lentiCRISPRv2 carrying both Cas9 enzyme and a gRNA transfected into HEK293T cells together with the packaging plasmids psPAX2 and pCMV-VSV-G (Addgene) at the ratio of 5:3:2. PKR: gRNA1: GATGGAAGAGAATTTCCAGA; gRNA2: AGTGTG-CATCGGGGGTGCAT; gRNA3: TGGTACAGGTTCTACTAAAC (ABM, 19075111). Two days after transfection, the supernatant containing lentiviruses was collected. Cells were infected by the mixed lentiviruses containing gRNA1-3. 36 h after infection, the cells were selected with puromycin (2 μg/mL) for one week in order to select knockout cells. Knockout cells were confirmed by western blot. Selection of single clones was performed by dilution in 96-well, which were confirmed by western blots.

## Generating G3BP1-GFP cell line

Transfected U2OS Flp-In cells (generated by Flp-In system, Thermo-Fisher) with G3BP1-GFP reconstructed plasmid and the pOG44 expression plasmid at ration of 9:1. 24 h after transfection, washed the cells and added fresh medium to the cells. 48 h after transfection, split the cells into fresh medium around 25% confluent. Incubate the cells at 37 °C for 2–3 h until they have attached to the culture dish. Then the medium was removed and added with fresh medium containing 100 μg/mL hygromycin. Cells were further fed with selective medium every 3–4 days until single cell clone can be identified. Picked hygromycin-resistant clones and expanded each clone to test.

## LysoIP assay

Lentiviruses constructs for generating stable LysoIP cells were purchased from Addgene. HEK293T cells were transfected with pLJC5-TMEM192-3xHA or pLJC5-TMEM192-2xFLAG constructs in combination with psPAX2 and pCMV-VSV-G packaging plasmids, at the ratio of 5:3:2, 60 h after transfection, the supernatant containing lentiviruses was collected and centrifuged to remove cells and then frozen at −80 °C. To establish LysoIP stably expressing cell lines, cells were plated in 10 cm dish in DMEM with 10% FBS and infected with 500 μL of virus-containing media overnight, then add puromycin for selection.

Selected cells in 15 cm plates with 90% confluency were used for each LysoIP. Cells with or without treatment were quickly rinsed twice with PBS and then scraped in 1 mL of KPBS (136 mM KCl, 10 mM KH$_2$PO$_4$, pH 7.25 was adjusted with KOH) and centrifuged at 3000 rpm for 2 min at 4 °C. Pelleted cells were resuspended in 950 μL KPBS and reserved 25 μL for further processing of the whole-cell lysate. The remaining cells were gently homogenized with 20 strokes of a 2 mL homogenizer. The homogenate was then centrifuged at 3000 rpm for 2 min at 4 °C and the supernatant was incubated with 100 μL of KPBS prewashed anti-HA magnetic beads (ThermoFisher) on a gentle rotator shaker for 15 min. Immunoprecipitants were then gently washed three times with KPBS and eluted with 2 x Laemmli sample buffer (Bio-Rad) and subjected to immunoblot analysis.

## High content microscopy (HCM) analysis

Cells in 96-well plates were fixed in 4% paraformaldehyde for 5 min. Cells were then permeabilized with 0.1% saponin in 3%

Bovine serum albumin (BSA) for 30 min followed by incubation with primary antibodies for 2 h and secondary antibodies for 1 h. The analysis of Poly(A) RNA involved diluting a stock of 5'-labeled Cy3-Oligo-dT(30) stock (GeneLink) to a final concentration of 1 ng/µL, and incubation at 37 °C for at least one hour. Hoechst 33342 staining was performed for 3 min. HCM with automated image acquisition and quantification was carried out using a Cellomics HCS scanner and iDEV software (ThermoFisher). Automated epifluorescence image collection was performed for a minimum of 500 cells per well. Epifluorescence images were machine analyzed using preset scanning parameters and object mask definitions. Hoechst 33342 staining was used for autofocus and to automatically define cellular outlines based on background staining of the cytoplasm. Primary objects were cells, and regions of interest (ROI) or targets were algorithm-defined by shape/segmentation, maximum/minimum average intensity, total area and total intensity, to automatically identify puncta or other profiles within valid primary objects. All data collection, processing (object, ROI, and target mask assignments) and analyses were computer driven independently of human operators. HCM provides variable statistics since it does not rely on parametric reporting cells as positive or negative for a certain marker above or below a puncta number threshold.

## PI uptake assay

20,000 cells were plated in each well of a 96-well plate. Subsequently, cells were treated with lysosomal damaging agents, such as LLOMe. PI (propidium iodide) uptake was measured after 5 min incubation with 100 µg/mL diluted PI solution (Thermo-Fisher) in complete medium at 37 °C. After PI incubation, cells were fixed with 4% paraformaldehyde and stained with Hoechst 33342 for HCM analysis.

## LDH release assay

Each well of a 96-well plate was initially plated with 20,000 cells. Cells were treated with lysosomal damaging agents as indicated. Following this, the supernatant was measured for LDH (Lactate dehydrogenase) release using the kit of CytoTox 96® Non-Radioactive Cytotoxicity Assay (Promega, G1780), according to the manufacturer's instructions.

## Amnis flow cytometry analysis

Cells after treatment were washed with 3% BSA in PBS supplemented with 0.1% of $NaN_3$ before staining. Cells were stained using LIVE/DEAD™ Fixable Green Dead Cell Stain Kit (ThermoFisher) following the manufacturer's instructions. After staining, cells were then resuspended with 3% BSA in PBS supplemented with 0.1% of $NaN_3$ until acquisition on Amins ImageStreamx MKII (ISx, EMD Millipore, Seattle, WA, USA).

## LysoTracker assay

LysoTracker (LTR) Staining Solution was prepared by freshly diluting 2 µL of LTR stock solution (1 mM LysoTracker Red DND-99; Sigma-Aldrich, L7528) in 1 mL of medium. 10 µL of Lyso-Tracker Staining Solution was added to 90 µL of medium each well

in 96-well plates (final volume 100 µL per well, final concentration 0.2 µM LTR) and adherent cells incubated at 37 °C for 30 min protected from light. Wells were rinsed gently by 1 × PBS and fixed in 4% paraformaldehyde for 2 min. Wells were washed once in 1 × PBS and nuclei stained with Hoechst 33342 for 2 min before analyzing the plates by HCM.

## Co-immunoprecipitation assay

Cells transfected with 8–10 µg of plasmids were lysed in NP-40 buffer (ThermoFisher) supplemented with protease inhibitor cocktail (Roche, 11697498001) and 1 mM PMSF (Sigma, 93482) for 30 min on ice. Supernatants were incubated with (2–3 µg) antibodies overnight at 4 °C. The immune complexes were captured with Dynabeads (ThermoFisher), followed by three times washing with 1 × PBS. Proteins bound to Dynabeads were eluted with 2 × Laemmli sample buffer (Bio-Rad) and subjected to immunoblot analysis. Immunoblotting images were visualized and analyzed using ImageLab v.6.0.0.

## GST pulldown assay

This assay was performed using the Pierce™ GST Protein Interaction Pull-Down Kit (Thermo, 21516) according to the manufacturer's instructions. Sample Preparation: Remove reduced glutathione from the previously purified protein sample by dialysis against TBS (BupH Tris Buffered Saline). Determine the protein concentration of the GST-tagged fusion protein sample using the BCA Protein Assay Kit (Thermo, 23227). Glutathione Agarose Preparation: Equilibrate Glutathione Agarose by washing multiple times with wash solution. Bait Protein Immobilization: Immobilize the bait protein (200 µg of GST-tagged protein per sample) for 2 h at 4 °C. $CaCl_2$ was added at a final concentration of 10 µM to activate ALG2 and ALIX. Prey Protein Capture: Add prey protein from previously purified samples, using ~100–150 µg of protein per sample. Incubate for prey protein capture overnight at 4 °C. Bait-Prey Elution: Elute the bait-prey complex using fresh elution buffer for each experiment. Protein Analysis: Add 4× SDS gel loading buffer to the eluted samples. Separate proteins by SDS-PAGE for detection of respective proteins.

## Immunofluorescence confocal microscopy analysis

Cells were plated onto coverslips in 6-well plates. After treatment, cells were fixed in 4% paraformaldehyde for 5 min followed by permeabilization with 0.1% saponin in 3% BSA for 30 min. Cells were then incubated with primary antibodies for 2 h and appropriate secondary antibodies Alexa Fluor 488 or 568 (Thermo-Fisher) for 1 h at room temperature. Coverslips were mounted using Prolong Gold Antifade Mountant (ThermoFisher). Images were acquired using a confocal microscope (META; Carl Zeiss) equipped with a 63 3/1.4 NA oil objective, camera (LSM META; Carl Zeiss), and AIM software (Carl Zeiss).

## APEX2-labeling and streptavidin enrichment for LC/MS/MS DIA analysis

HEK293T cells transfected APEX2 - eIF2α were incubated with 1 mM LLOMe for 1 h (confluence of cells remained at 70–80%). Cells were next incubated in 500 mM biotin-phenol (AdipoGen) for

the last 45 min of LLOMe incubation. A 1 min pulse with 1 mM $H_2O_2$ at room temperature was stopped with quenching buffer (10 mM sodium ascorbate, 10 mM sodium azide and 5 mM Trolox in Dulbecco's Phosphate Buffered Saline (DPBS)). All samples were washed twice with quenching buffer, and twice with DPBS.

For mass spectrometry analysis, cell pellets were lysed in 500 mL ice-cold lysis buffer (6 M urea, 0.3 M NaCl, 1 mM EDTA, 1 mM EGTA, 10 mM sodium ascorbate, 10 mM sodium azide, 5 mM Trolox, 1%glycerol and 25 mm Tris/HCl, PH 7.5) for 30 min by gentle pipetting. Lysates were clarified by centrifugation and protein concentrations determined as above. Streptavidin-coated magnetic beads (Pierce) were washed with lysis buffer. 3 mg of each sample was mixed with 100 mL of streptavidin bead. The suspensions were gently rotated at 4 °C for overnight to bind biotinylated proteins. The flowthrough after enrichment was removed and the beads were washed in sequence with 1 mL IP buffer (150 mM NaCl, 10 mM Tris-HCl pH 8.0, 1 mM EDTA, 1 mM EGTA, 1% Triton X-100) twice; 1 mL 1 M KCl; 1 mL of 50 mM $Na_2CO_3$; 1 mL 2 M Urea in 20 mM Tris HCl pH 8; 1 mL IP buffer. Biotinylated proteins were eluted, 10% of the sample processed for Western Blot and 90% of the sample processed for LC/MS/MS DIA (data-independent acquisition mass spectrometry) analysis.

LC/MS/MS DIA were performed at UC Davis Proteomics Core Facility (Davis, CA). Protein samples on magnetic beads were washed four times with 200 μL of 50 mM triethyl ammonium bicarbonate (TEAB) with a 20 min shake time at 4 °C in between each wash. Roughly 2.5 mg of trypsin was added to the bead and TEAB mixture and the samples were digested overnight at 800 rpm shake speed. After overnight digestion the supernatant was removed, and the beads were washed once with enough 50 mM ammonium bicarbonate to cover. After 20 min at a gentle shake the wash is removed and combined with the initial supernatant. The peptide extracts are reduced in volume by vacuum centrifugation and a small portion of the extract is used for fluorometric peptide quantification (ThermoFisher). One microgram of sample based on the fluorometric peptide assay was loaded for each LC/MS/MS analysis.

Peptides were separated on an Easy-spray 100 mm × 25 cm C18 column using a Dionex Ultimate 3000 nUPLC. Solvent A = 0.1% formic acid, Solvent B = 100% Acetonitrile 0.1% formic acid. Gradient conditions = 2% B to 50% B over 60 min, followed by a 50–99% B in 6 min and then held for 3 min than 99% B to 2% B in 2 min. Total Run time = 90 min. Thermo Scientific Fusion Lumos mass spectrometer running in data-independent analysis (DIA) mode. Two gas phases fractionated (GFP) injections were made per sample using sequential 4 Da isolation widows. GFP1 = $m/z$ 362–758, GFP 2 = $m/z$ 758–1158. Tandem mass spectra were acquired using a collision energy of 30, resolution of 30 K, maximum inject time of 54 ms and a AGC target of 50 K.

### DIA quantification and statistical analysis

DIA data was analyzed using Spectronaut. Raw data files were converted to mzML format using ProteoWizard (3.0.11748). Analytic samples were aligned based on retention times and individually searched against Pan human library http://www.swathatlas.org/ with a peptide mass tolerance of 10.0 ppm and a fragment mass tolerance of 10.0 ppm. Variable modifications considered were: Modification on M M and Modification on C C. The digestion enzyme was assumed to be Trypsin with a maximum

of 1 missed cleavage site(s) allowed. Only peptides with charges in the range <2..3> and length in the range <6..30> were considered. Peptides identified in each sample were filtered by Percolator (3.01.nightly-13-655e4c7-dirty) to achieve a maximum FDR of 0.01. Individual search results were combined, and peptide identifications were assigned posterior error probabilities and filtered to an FDR threshold of 0.01 by Percolator (3.01.nightly-13-655e4c7-dirty). Peptide quantification was performed by Encyclopedia (0.8.1). For each peptide, the 5 highest quality fragment ions were selected for quantitation. Proteins that contained similar peptides and could not be differentiated based on MS/MS analysis were grouped to satisfy the principles of parsimony. Proteins with a minimum of 2 identified peptides were thresholded to achieve a protein FDR threshold of 1.0%. Raw data and Spectronaut results are in Dataset EV1.

### Quantification and statistical analysis

Data in this study are presented as means ± SEM ($n ≥ 3$). Data were analyzed with either analysis of variance (ANOVA) with Tukey's HSD post hoc test, or a two-tailed Student's t test. For HCM, $n ≥ 3$ includes in each independent experiment: 500 valid primary objects/cells per well, from ≥5 wells per plate per sample. Quantification of immunoblotting based on three independent experiments. Band intensities were quantified using ImageJ software. Results are presented as mean ± SEM. Statistical significance was determined using ANOVA, with $p < 0.05$ considered statistically significant. Statistical significance was defined as: † (not significant) $p ≥ 0.05$ and $*p < 0.05$, $**p < 0.01$.

## Data availability

Raw MS DIA data of APEX2 - eIF2α in HEK293T cells have been deposited in the MassIVE proteomics repository (Dataset ID: MSV000093768) and in ProteomeXchange (Dataset ID: PXD048258). Source data is deposited in BioStudies, access number: S-BSST1652.

The source data of this paper are collected in the following database record: biostudies:S-SCDT-10_1038-S44318-024-00292-1.

## Peer review information

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

## Acknowledgements

We thank Dr. Rakez Kayed (The University of Texas Medical Branch, Galveston, TX) for tau aggregates. We thank Dr. Carmen San Martín (Centro Nacional de Biotecnología, Spain) for human adenovirus mutant TS1. This work was supported by an NIH AIM COBRE grant (P20GM121176) and an NIGMS R35 grant (R35GM154651) to J Jia. Mass spectrometry analysis was supported by an NIH S10 grant (S10OD026918-01A1) to B Phinney. The human adenovirus studies were supported by NIH RF1 grants (R01EY021558 and R01EY013124) to J Rajaiya. The hemozoin studies were supported by an NIH R01 grant (AI130473) to D Perkins. The tau studies were supported by an NIH RF1 grant (RF1NS083704) to K Bhaskar. AH Lystad was supported by a Young Research Talents Grant from the Research Council of Norway (Project number 325305). J Pu was supported by an NIGMS R35 grant (R35GM147419). M Rosas-Lemus was support by an NIH AIM COBRE grant (P20GM121176).

## Author contributions

**Jacob Duran**: Data curation; Formal analysis; Investigation; Methodology. **Jay E Salinas**: Formal analysis; Investigation; Methodology; Writing—review and editing. **Rui ping Wheaton**: Formal analysis; Investigation; Methodology; Writing—review and editing. **Suttinee Poolsup**: Data curation; Formal analysis; Investigation; Methodology. **Lee Allers**: Software; Formal analysis. **Monica Rosas-Lemus**: Software. **Li Chen**: Software. **Qiuying Cheng**: Resources. **Jing Pu**: Software. **Michelle Salemi**: Formal analysis; Investigation. **Brett Phinney**: Formal analysis. **Pavel Ivanov**: Resources. **Alf Håkon Lystad**: Investigation; Writing—review and editing. **Kiran Bhaskar**: Resources. **Jaya Rajaiya**: Resources. **Douglas J Perkins**: Resources; Writing—review and editing. **Jingyue Jia**: Conceptualization; Resources; Data curation; Software; Formal analysis; Supervision; Funding acquisition; Validation; Investigation; Visualization; Methodology; Writing—original draft; Project administration; Writing—review and editing.

Source data underlying figure panels in this paper may have individual authorship assigned. Where available, figure panel/source data authorship is

listed in the following database record: biostudies:S-SCDT-10_1038-S44318-024-00292-1.

## Disclosure and competing interests statement

The authors declare no competing interests.

# Expanded View Figures

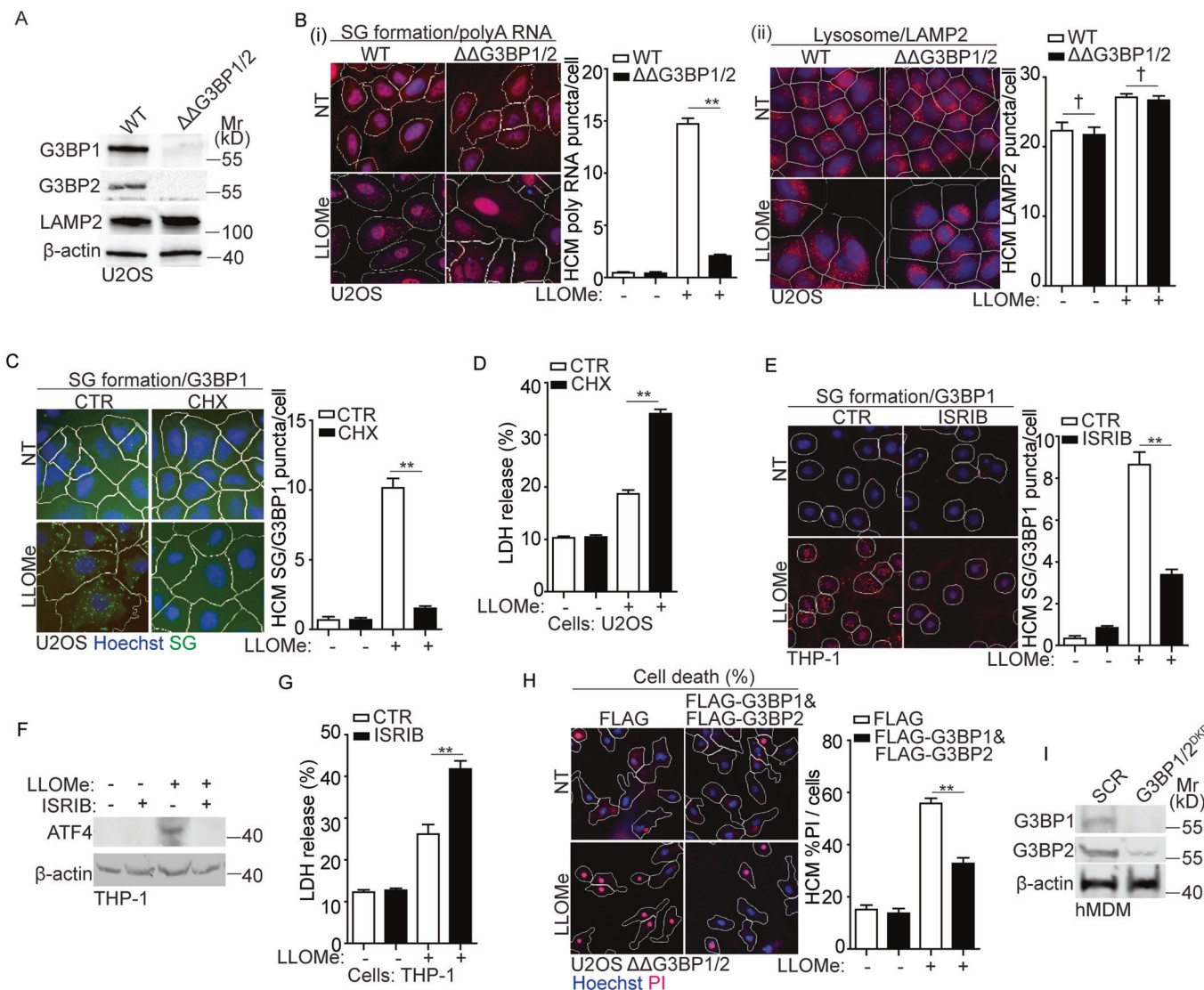

**Figure EV1. Stress granule formation is important for cell survival during lysosomal damage.**

(A) Immunoblot analysis of G3BP1 and G3BP2 in U2OS WT and ΔΔG3BP1/2 cells. (B) Quantification by high-content microscopy (HCM) of polyA RNA (Cy3-oligo[dT]) by FISH (i) and LAMP2 (ii) in U2OS WT and ΔΔG3BP1/2 cells. Cells were treated with 2 mM LLOMe for 30 min. White masks, algorithm-defined cell boundaries (primary objects); red masks, computer-identified polyA RNA or LAMP2 puncta respectively (target objects). (C) Quantification by HCM of G3BP1 puncta in U2OS cells. Cells were treated with 2 mM LLOMe in the presence or absence of 10 μg/ml cycloheximide (CHX) for 30 min. White masks, algorithm-defined cell boundaries; green masks, computer-identified G3BP1 puncta. (D) Cell death analysis of supernatants of U2OS cells by a LDH release assay. Cells were treated with 2 mM LLOMe in the presence or absence of 10 μg/ml CHX for 30 min. (E) Quantification by HCM of G3BP1 puncta in human monocytic THP-1 cells. Cells were treated with 1 mM LLOMe in the presence or absence of 200 nM ISRIB for 30 min. White masks, algorithm-defined cell boundaries; red masks, computer-identified G3BP1 puncta. (F) Immunoblot analysis of ATF4 in THP-1 cells treated with 1 mM LLOMe in the presence or absence of 200 nM ISRIB for 30 min. (G) Cell death analysis of supernatants of THP-1 cells by a LDH release assay. Cells were treated with 1 mM LLOMe in the presence or absence of 200 nM ISRIB for 30 min. (H) Quantification of cell death by HCM using a propidium iodide (PI) uptake assay in U2OS G3BP1&2 double knockout (ΔΔG3BP1/2) cells overexpressing either FLAG or FLAG-G3BP1 & FLAG-G3BP2. Cells were treated with 2 mM LLOMe for 30 min, and then stained with propidium iodide (PI) (dead cells) and Hoechst-33342 (total cells). White masks, algorithm-defined cell boundaries; red masks, computer-identified PI+ nuclei. (I) Immunoblot analysis of the protein level of G3BP1 and G3BP2 in hMDM transfected with scrambled siRNA as control (SCR) or G3BP1 and G3BP2 siRNA for double knockdown (DKD). CTR, control; NT, untreated cells. Data, means ± SEM ($n = 3$); HCM: $n \geq 3$ (each experiment: 500 valid primary objects/cells per well, ≥5 wells/sample). †$p \geq 0.05$ (not significant), **$p < 0.01$, ANOVA. See also Fig. 1.

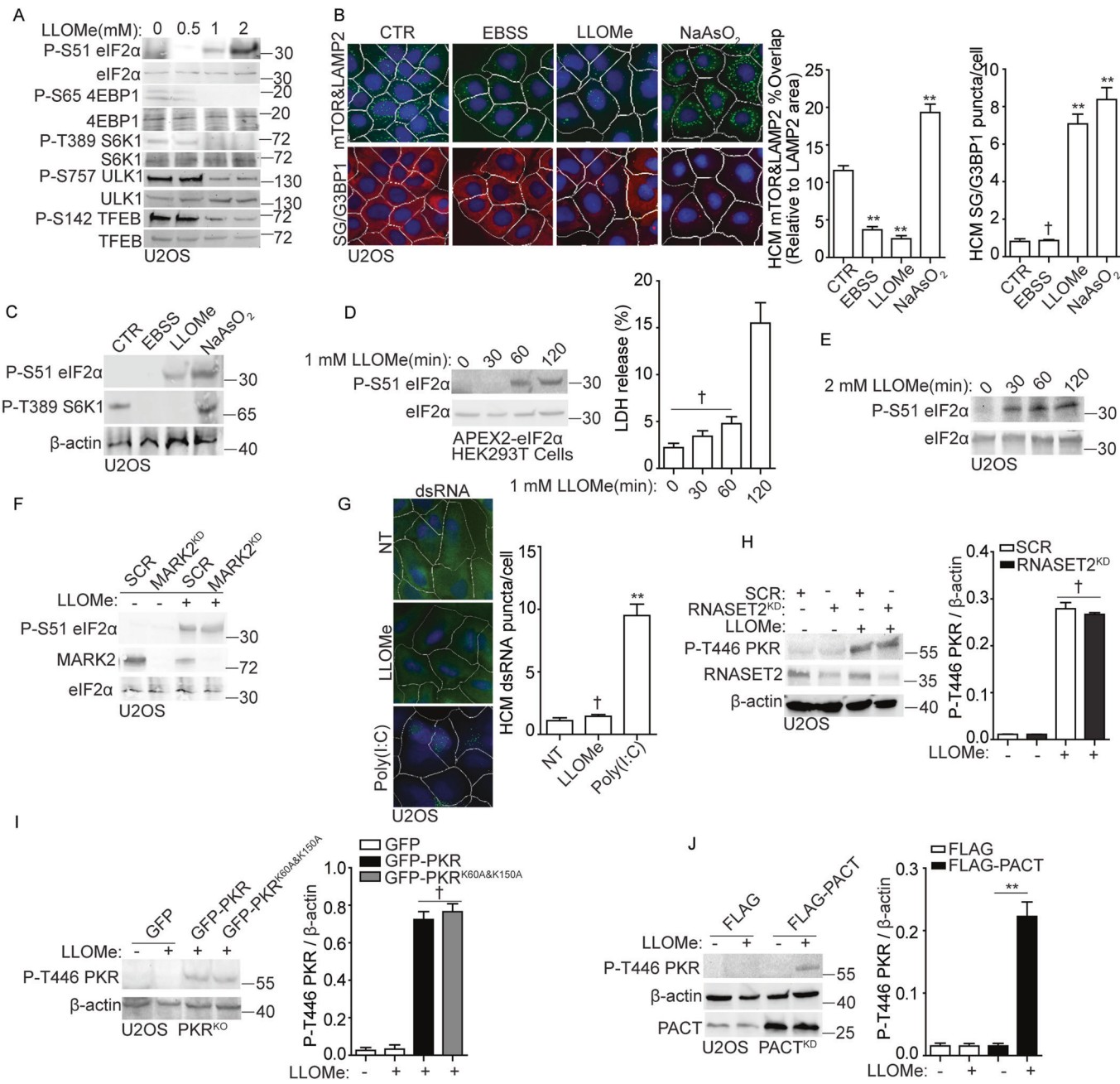

**Figure EV2.  PACT-PKR- eIF2α pathway controls stress granule formation in response to lysosomal damage.**

(**A**) Immunoblot analysis of phosphorylation of eIF2α (S51), 4EBP1 (Ser65), S6K (Thr389), ULK1 (Ser757) and TFEB (Ser142) in U2OS cells treated with the indicated dose of LLOMe for 30 min. (**B**) Quantification by HCM of overlaps between mTOR and LAMP2 or G3BP1 puncta in U2OS cells. Cells were treated with EBSS, 2 mM LLOMe or 100 μM NaAsO2 for 30 min. White masks, algorithm-defined cell boundaries; green masks, computer-identified overlap between mTOR and LAMP2; red masks, computer-identified G3BP1 puncta. (**C**) Immunoblot analysis of phosphorylation of eIF2α (S51) and S6K1 (T389) in U2OS cells treated as in (**B**). (**D**) Immunoblot analysis of phosphorylation of eIF2α (S51) and cell death analysis by a LDH release assay in HEK293T cells expressing APEX2-eIF2α. Cells were treated with 1 mM LLOMe for the indicated durations. (**E**) Immunoblot analysis of phosphorylation of eIF2α (S51) in U2OS cells. Cells were treated with 2 mM LLOMe for the indicated durations. (**F**) Immunoblot analysis of phosphorylation of eIF2α (S51) in U2OS cells transfected with either scrambled siRNA as control (SCR) or MARK2 siRNA for knockdown (MARK2$^{KD}$). Cells were treated with 2 mM LLOMe for 30 min. (**G**) Quantification by HCM of dsRNA puncta in U2OS cells. Cells were treated with 2 mM LLOMe or 100 ng/mL Poly (I:C) for 30 min. Green masks, computer-identified dsRNA puncta. (**H**) Immunoblot analysis of phosphorylation of PKR (T446) in U2OS cells transfected with either scrambled siRNA as control (SCR) or RNASET2 siRNA for knockdown (RNASET2$^{KD}$). Cells were treated with 2 mM LLOMe for 30 min. The level of phosphorylation of PKR (T446) was quantified based on three independent experiments. (**I**) Immunoblot analysis of phosphorylation of PKR (T446) in PKR$^{KO}$ U2OS G3BP1-GFP cells, overexpressing GFP, GFP-PKR and GFP-PKR$^{K60A\&K150A}$. Cells were treated with 2 mM LLOMe for 30 min. The level of phosphorylation of PKR (T446) was quantified based on three independent experiments. (**J**) Immunoblot analysis of phosphorylation of PKR (T446) in U2OS PACT knockdown cells (PACT$^{KD}$) overexpressing FLAG or FLAG-PACT. Cells were treated with 2 mM LLOMe for 30 min. The level of phosphorylation of PKR (T446) was quantified based on three independent experiments. CTR, control. Data, means ± SEM ($n = 3$); HCM: $n \geq 3$ (each experiment: 500 valid primary objects/cells per well, ≥5 wells/sample). †$p \geq 0.05$ (not significant), **$p < 0.01$, ANOVA. See also Figs. 2 and 3.

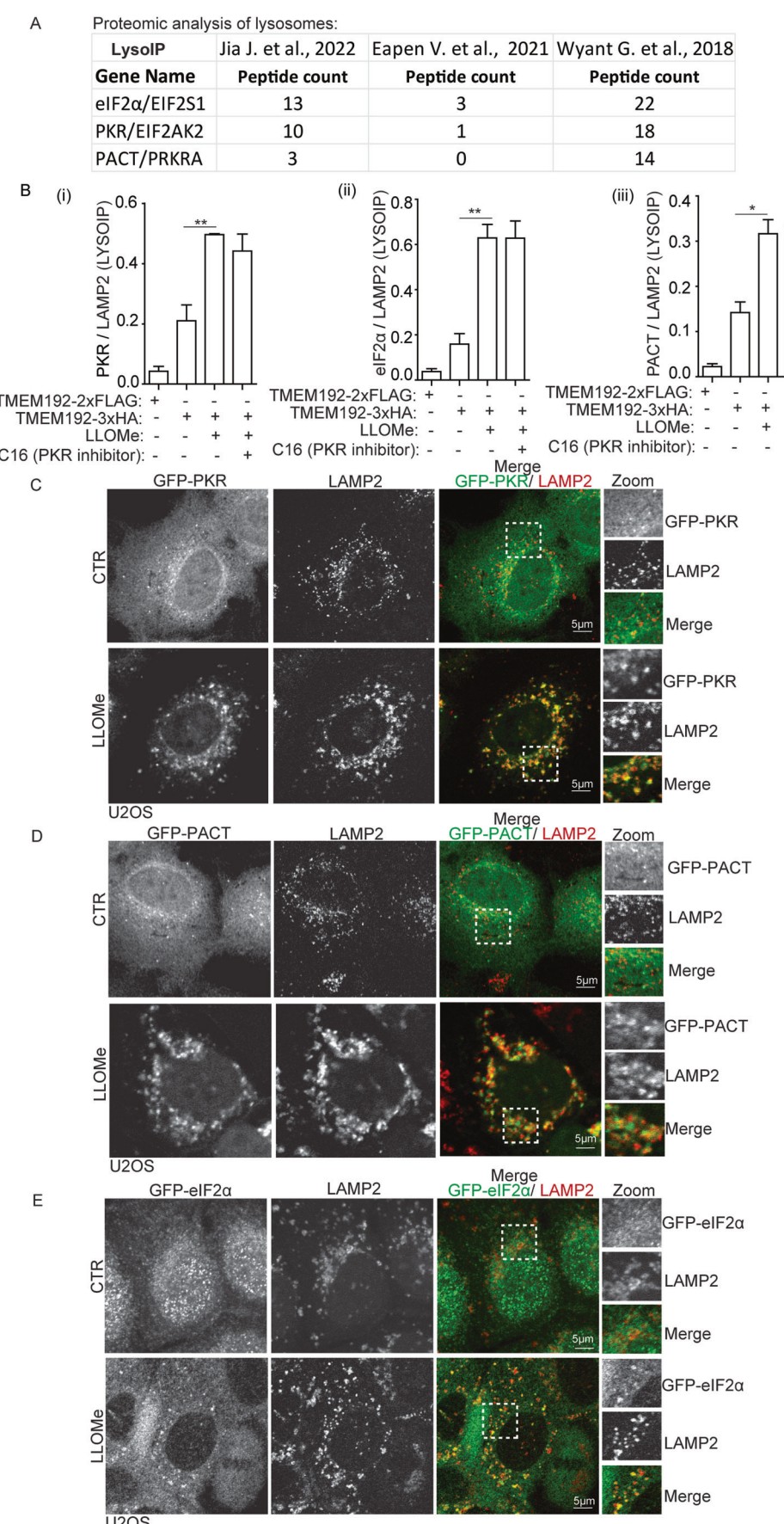

A   Proteomic analysis of lysosomes:

| LysoIP | Jia J. et al., 2022 | Eapen V. et al., 2021 | Wyant G. et al., 2018 |
|---|---|---|---|
| Gene Name | Peptide count | Peptide count | Peptide count |
| eIF2α/EIF2S1 | 13 | 3 | 22 |
| PKR/EIF2AK2 | 10 | 1 | 18 |
| PACT/PRKRA | 3 | 0 | 14 |

◀ **Figure EV3. PKR, PACT and eIF2α are associated with damaged lysosomes.**

(A) Summary of the literature on the detected peptide count of PKR, PACT and eIF2α in the proteomic analysis of lysosomes based on LysoIP LC/MS/MS analysis. (B) Quantification of Fig. 3F; the level of PKR, eIF2α and PACT in LysoIP was quantified based on three independent experiments. (C) Confocal microscopy imaging of GFP-PKR and LAMP2 in U2OS cells treated with 2 mM LLOMe for 30 min. Scale bar, 5 µm. (D) Confocal microscopy imaging of GFP-PACT and LAMP2 in U2OS cells treated with 2 mM LLOMe for 30 min. Scale bar, 5 µm. (E) Confocal microscopy imaging of GFP-eIF2α and LAMP2 in U2OS cells treated with 2 mM LLOMe for 30 min. Scale bar, 5 µm. *$p < 0.05$, **$p < 0.01$, ANOVA. See also Fig. 3.

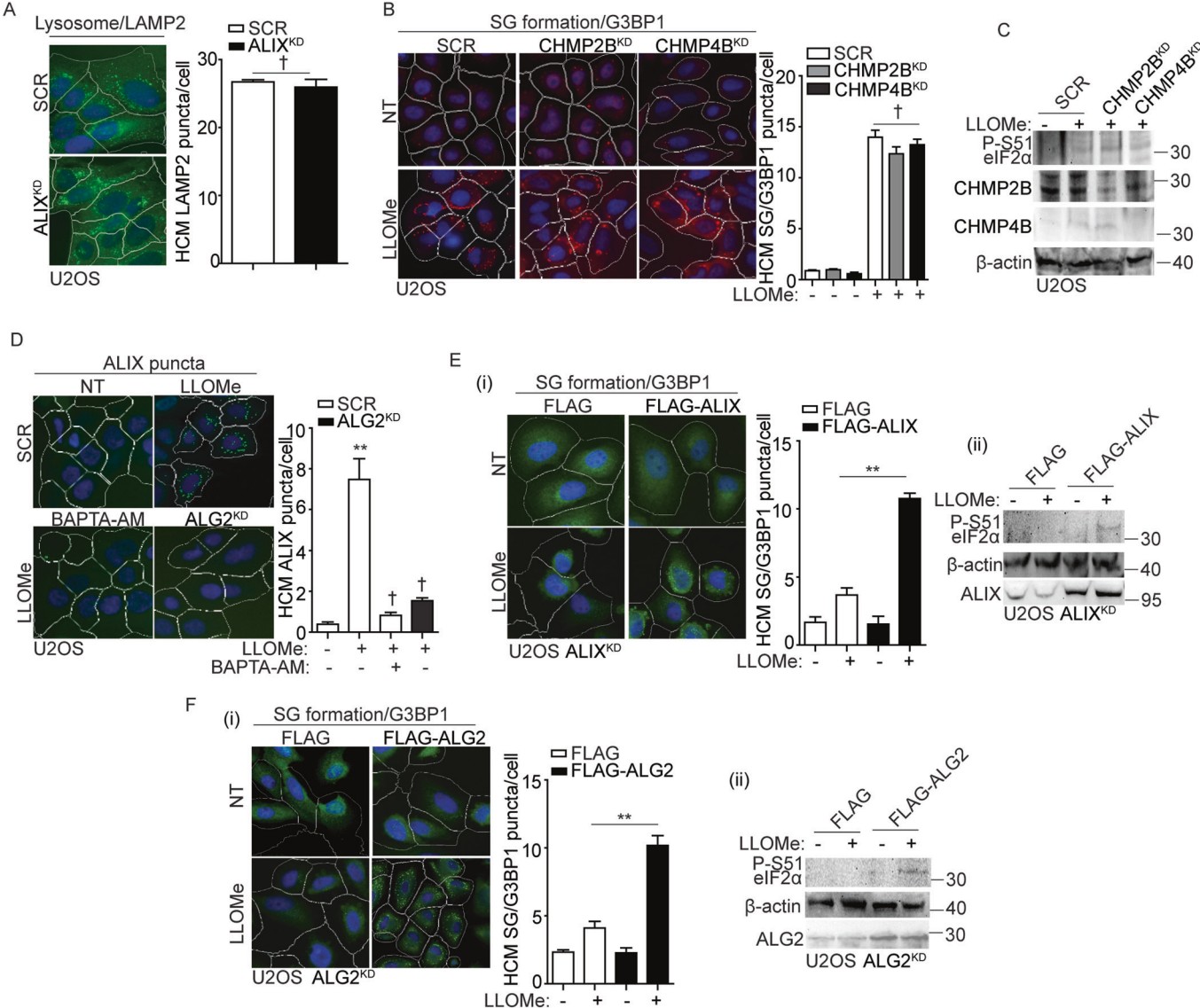

**Figure EV4.  ALIX regulates stress granule formation during lysosomal damage.**

(A) Quantification by HCM of LAMP2 in U2OS cells transfected with scrambled siRNA as control (SCR), or ALIX siRNA for knockdown (ALIX^KD). White masks, algorithm-defined cell boundaries; green masks, computer-identified LAMP2 puncta. (B) Quantification by HCM of G3BP1 puncta in U2OS cells transfected with scrambled siRNA as control (SCR), CHMP2B siRNA for knockdown (CHMP2B^KD) or CHMP4B siRNA for knockdown (CHMP4B^KD). Cells were treated with 2 mM LLOMe for 30 min. White masks, algorithm-defined cell boundaries; red masks, computer-identified G3BP1 puncta. (C) Immunoblot analysis of phosphorylation of eIF2α (S51) in U2OS transfected with scrambled siRNA as control (SCR), CHMP2B siRNA for knockdown (CHMP2B^KD) or CHMP4B siRNA for knockdown (CHMP4B^KD), subjected to 2 mM LLOMe treatment for 30 min. (D) Quantification by HCM of ALIX puncta in U2OS cells transfected with scrambled siRNA as control (SCR), or ALG2 siRNA for knockdown (ALG2^KD), or pre-treated with 15 μM BAPTA-AM for 1 h. Cells were treated with 2 mM LLOMe for 30 min. White masks, algorithm-defined cell boundaries; green masks, computer-identified ALIX puncta. (E) (i) Quantification by HCM of G3BP1 puncta in U2OS ALIX knockdown cells (ALIX^KD) overexpressing FLAG or FLAG-ALIX. Cells were treated with 2 mM LLOMe for 30 min. White masks, algorithm-defined cell boundaries; green masks, computer-identified G3BP1 puncta. (ii) Immunoblot analysis of phosphorylation of eIF2α (S51) in U2OS cells as described in (i). (F) (i) Quantification by HCM of G3BP1 puncta in U2OS ALG2 knockdown cells (ALG2^KD) overexpressing FLAG or FLAG-ALG2. Cells were treated with 2 mM LLOMe for 30 min. White masks, algorithm-defined cell boundaries; green masks, computer-identified G3BP1 puncta. (ii) Immunoblot analysis of phosphorylation of eIF2α (S51) in U2OS cells as described in (i). NT, untreated cells. Data, means ± SEM ($n = 3$); HCM: $n \geq 3$ (each experiment: 500 valid primary objects/cells per well, ≥5 wells/sample). †$p \geq 0.05$ (not significant), **$p < 0.01$, ANOVA. See also Fig. 4.

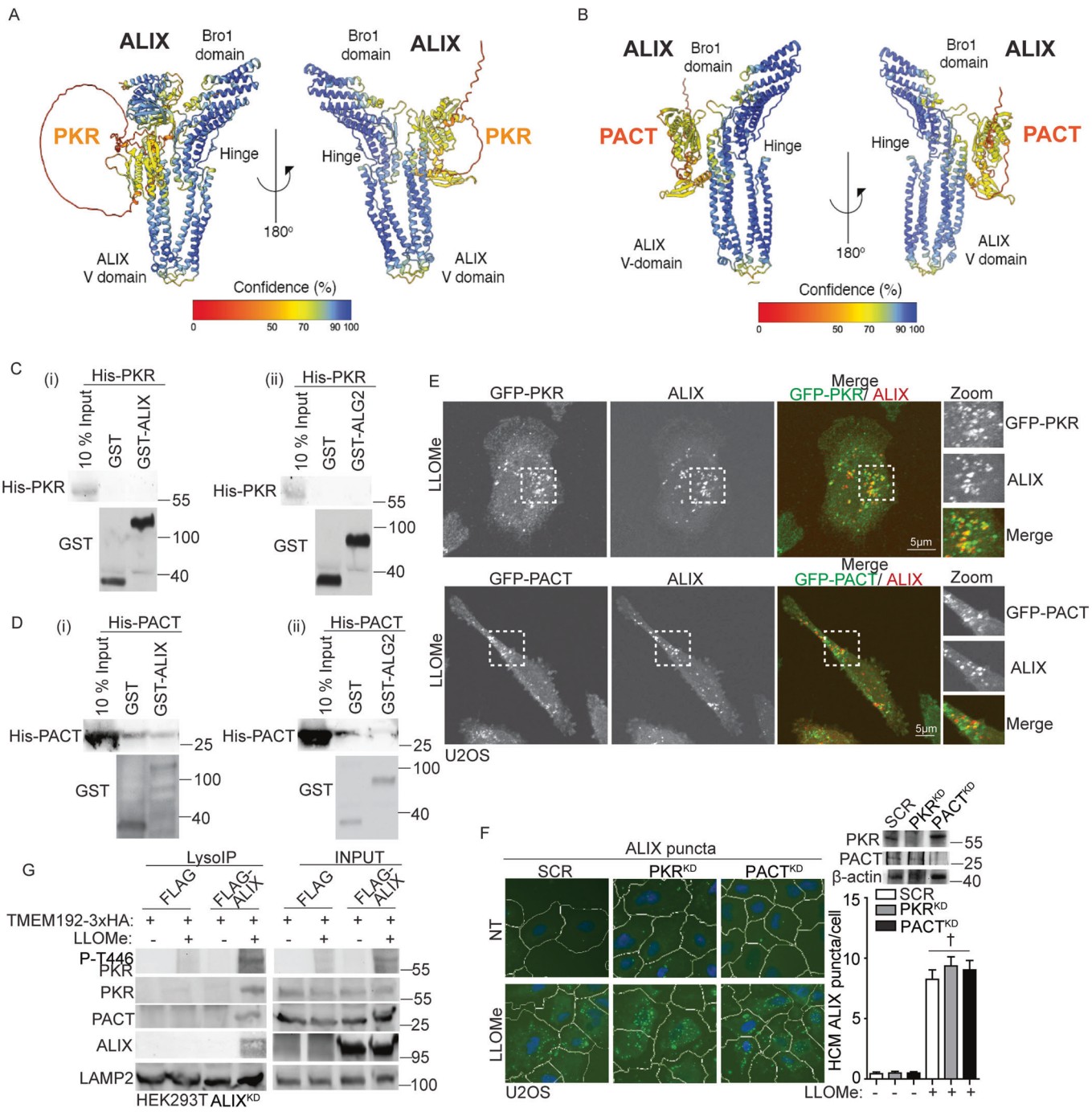

**Figure EV5. PKR and PACT associate with ALIX during lysosomal damage.**

(A) AlphaFold 2 predicted the interaction between PKR and ALIX, with the C-terminal PRD domain removed. (B) AlphaFold 2 predicted the interaction between PACT and ALIX, with the C-terminal PRD domain removed. (C) GST pulldown assay of in vitro translated His-tagged PKR with GST or GST-tagged ALIX (i) or ALG2 (ii) in the presence of 10 μM CaCl2. (D) GST pulldown assay of in vitro translated His-tagged PACT with GST or GST-tagged ALIX (i) or ALG2 (ii) in the presence of 10 μM CaCl2. (E) Confocal microscopy imaging of GFP-PKR/PACT and ALIX in U2OS cells treated with 2 mM LLOMe for 30 min. Scale bar, 5 μm. (F) Quantification by HCM of ALIX puncta in U2OS cells transfected with scrambled siRNA as control (SCR), PKR siRNA for knockdown (PKRKD), or PACT siRNA for knockdown (PACTKD). Cells were treated with 2 mM LLOMe for 30 min. White masks, algorithm-defined cell boundaries; green masks, computer-identified ALIX puncta. (G) Analysis of proteins associated with purified lysosomes (LysoIP; TMEM192-3xHA) from HEK293T ALIX knockdown cells (ALIXKD) overexpressing FLAG or FLAG-ALIX. Cells were treated with 1 mM LLOMe for 1 h. NT, untreated cells. Data, means ± SEM ($n = 3$); HCM: $n \geq 3$ (each experiment: 500 valid primary objects/cells per well, ≥5 wells/sample). †$p \geq 0.05$ (not significant), ANOVA. See also Fig. 5.

