## [Peer Review File · The EMBO Journal]

Calcium signaling from damaged lysosomes induces cytoprotective stress granules

Jacob duran, Jay Salinas, Rui Wheaton, Suttinee Poolsup, Lee Allers, Monica Leums, Li chen, Qiuying Chen, Jing Pu, Michelle R. Salemi, Brett S. Phinney, pavel ivanov, Alf Lystad, Kiran Bhaskar, Jaya Rajaiya, Douglas Perkins, and Jingyue Jla

Corresponding author(s): Jingyue Jla (jjia@salud.unm.edu)

Review Timeline:

Submission Date:	20th May 24
Editorial Decision:	26th Jun 24
Revision Received:	18th Sep 24
Accepted:	11th Oct 24

Editor: William Teale

Transaction Report:

Dear Dr. Jia,

Thank you again for the submission of your manuscript entitled "A mechanism that transduces lysosomal damage signals to stress granule formation for cell survival" and for your patience during the review process. We have now received the reports from two referees, which I copy below.

As you can see from their comments, while referee #2 asks for some important clarifications and controls and referee #1 contributes some very helpful suggestions for how the work should be completed, both found it well done and timely. That said, they also list concerns that will require your attention before your manuscript can be published in The EMBO Journal.

Based on the overall interest expressed in the reports, I would like to invite you to address the comments of all referees in a revised version of the manuscript. I should add that it is The EMBO Journal policy to allow only a single major round of revision and that it is therefore important to resolve the main concerns at this stage. I believe the concerns of the referees are reasonable and addressable, but please contact me if you have any questions, need further input on the referee comments or if you anticipate any problems in addressing any of their points. I would be happy to have a Zoom call with you about the best way to proceed when you have had a chance to digest the referee reports. Please, follow the instructions below when preparing your manuscript for resubmission.

I would also like to point out that as a matter of policy, competing manuscripts published during this period will not be taken into consideration in our assessment of the novelty presented by your study ("scooping" protection). We have extended this 'scooping protection policy' beyond the usual 3 month revision timeline to cover the period required for a full revision to address the essential experimental issues. Please contact me if you see a paper with related content published elsewhere to discuss the appropriate course of action.

Again, please contact me at any time during revision if you need any help or have further questions.

Thank you very much again for the opportunity to consider your work for publication. I look forward to your revision.

Best regards,

William

William Teale, Ph.D.
Editor
The EMBO Journal

When submitting your revised manuscript, please carefully review the instructions below and include the following items:

- 1) a .docx formatted version of the manuscript text (including legends for main figures, EV figures and tables). Please make sure that the changes are highlighted to be clearly visible.
- 2) individual production quality figure files as .eps, .tif, .jpg (one file per figure).
- 3) a .docx formatted letter INCLUDING the reviewers' reports and your detailed point-by-point response to their comments. As part of the EMBO Press transparent editorial process, the point-by-point response is part of the Review Process File (RPF), which will be published alongside your paper.
- 4) a complete author checklist, which you can download from our author guidelines ([https://wol-prod-cdn.literatumonline.com/pb-assets/embo-site/Author Checklist%20-%20EMBO%20J-1561436015657.xlsx](https://wol-prod-cdn.literatumonline.com/pb-assets/embo-site/Author%20Checklist%20-%20EMBO%20J-1561436015657.xlsx)). Please insert information in the checklist that is also reflected in the manuscript. The completed author checklist will also be part of the RPF.
- 5) Please note that all corresponding authors are required to supply an ORCID ID for their name upon submission of a revised manuscript.
- 6) We require a 'Data Availability' section after the Materials and Methods. Before submitting your revision, primary datasets produced in this study need to be deposited in an appropriate public database, and the accession numbers and database listed

under 'Data Availability'. Please remember to provide a reviewer password if the datasets are not yet public (see <https://www.embopress.org/page/journal/14602075/authorguide#datadeposition>). If no data deposition in external databases is needed for this paper, please then state in this section: This study includes no data deposited in external repositories. Note that the Data Availability Section is restricted to new primary data that are part of this study.

Note - All links should resolve to a page where the data can be accessed.

8) For data quantification: please specify the name of the statistical test used to generate error bars and P values, the number (n) of independent experiments (specify technical or biological replicates) underlying each data point and the test used to calculate p-values in each figure legend. The figure legends should contain a basic description of n, P and the test applied. Graphs must include a description of the bars and the error bars (s.d., s.e.m.).

9) We would also encourage you to include the source data for figure panels that show essential data. Numerical data can be provided as individual .xls or .csv files (including a tab describing the data). For 'blots' or microscopy, uncropped images should be submitted (using a zip archive or a single pdf per main figure if multiple images need to be supplied for one panel). Additional information on source data and instruction on how to label the files are available at .

10) We replaced Supplementary Information with Expanded View (EV) Figures and Tables that are collapsible/expandable online (see examples in <https://www.embopress.org/doi/10.15252/embj.201695874>). A maximum of 5 EV Figures can be typeset. EV Figures should be cited as 'Figure EV1, Figure EV2" etc. in the text and their respective legends should be included in the main text after the legends of regular figures.

12) Our journal encourages inclusion of *data citations in the reference list* to directly cite datasets that were re-used and obtained from public databases. Data citations in the article text are distinct from normal bibliographical citations and should directly link to the database records from which the data can be accessed. In the main text, data citations are formatted as follows: "Data ref: Smith et al, 2001" or "Data ref: NCBI Sequence Read Archive PRJNA342805, 2017". In the Reference list, data citations must be labeled with "[DATASET]". A data reference must provide the database name, accession number/identifiers and a resolvable link to the landing page from which the data can be accessed at the end of the reference. Further instructions are available at .

When assembling figures, please refer to our figure preparation guideline in order to ensure proper formatting and readability in

print as well as on screen:

We realize that it is difficult to revise to a specific deadline. In the interest of protecting the conceptual advance provided by the work, we recommend a revision within 3 months (24th Sep 2024). Please discuss the revision progress ahead of this time with the editor if you require more time to complete the revisions. Use the link below to submit your revision:

Referee #1:

Duran and colleagues investigate the pathway leading to activation of the integrated stress response to lysosomal damage, and the role of the stress granules that result. The authors build on their previous work in which they identified the phosphorylation of the translation initiation factor eIF2 α by the stress kinase PKR and resulting translational repression upon treatment with lysosome-damaging agents. Here the authors elucidate the molecular details of PKR recruitment to damaged lysosomes via ALIX, an ESCORT machinery protein, and identify PACT as a positive regulator of PKR activity under these conditions. The precise mechanism by which PKR is activated remains to be elucidated. The results also indicate that stress granules, which form as a consequence of translational repression when lysosomes are damaged, contribute to improved cell survival. This has also been observed in the context of various pathogen infections.

The experiments are convincing and carefully controlled. Although the high-content microscopy images are quantified, the absence of a quantitative graph for all blots makes it difficult to visually understand the meaning and validity of the claims. Overall, with many novel findings, this work makes an important contribution to understanding the role of the integrated stress response and stress granules in cell survival, as well as the complex interplay between integrated stress responses and other damage response pathways. The main comments below should be addressed experimentally to reinforce certain conclusions.

The missing piece of evidence concerns the triggering of PKR activation. Does PKR detect cellular dsRNA? Is there an accumulation of dsRNA in the cytoplasm after LLOMe treatment, e.g. detectable by immunofluorescence using the J2 antibody? One way of answering this question in part is, as done in Figure 3 for PKR wt, to knockdown PKR and reconstitute cells with a PKR dsRNA-binding mutant (e.g. K60A and K150A, described in PMID: 7852324 and PMID: 8810342).

The authors use G3BP1/2 double KO cells to measure the impact of stress granules on cell survival during LLOMe treatment and observe higher survival in cells unable to form SGs. A similar result was obtained in the presence of CHX or ISRIB. However, although useful, these approaches remain indirect. In recent years, it has been shown that G3BP1 is a multifunctional protein, also involved in ribosome quality control, PKR regulation, etc. On the other hand, the cycloheximide and ISRIB interfere at different levels with translation. Direct proof could be provided by the use of G3BP small-molecule inhibitors FAZ3532 or FAZ3780 recently published by the Taylor laboratory (PMID: 38284934) and now commercially available. In an identical context

of translation inhibition and in the presence of G3BP1, they would demonstrate that stress granule assembly itself is necessary for cell survival.

Figure 6B: p-PKR is not visibly detectable in the LLOMe-treated U2OS control (SCR), whereas it is in Figures 3E and 4B. I think the choice of blot is not the best. The same applies to figure 6C. IP staining for PACT looks "dubious" due to the absence of staining in lanes 2 to 4. A better choice of images would avoid this impression, but overall, as mentioned above, quantification of all blots is essential.

Figure 7 is very interesting, linking lysosomal damage to contexts more relevant than that of LLOMe use. However, the conclusions are somewhat preliminary at this stage, as none of the results provide information on the involvement of PKR and PACT in these processes. Is PKR phosphorylated during these different treatments? Figure 7B and S6D show only eIF2 α phosphorylation. Can these results on survival in the various pathological situations be reversed by the C16 inhibitor?

Referee #2:

This manuscript describes "a mechanism that transduces lysosomal damage signals to stress granule formation for cell survival." The authors use a combination of cell biological and biochemical methods to demonstrate that ALIX and ALG2 sense calcium leakage from damaged lysosomes. This leads to the activation of ALIX and the association of PKR with its activator PACT. Activated PKR phosphorylates eIF2 α , which in turn leads to the G3BP1/2-dependent formation of stress granules (SGs). The activation of this pathway, induced by lysosomal damage and the formation of SGs, is important for cell survival in response to diverse pathogenic challenges (hemozoin, adenoviral infections) associated with major human diseases. The authors delineate (supported by data) the pathway leading from lysosomal damage to SG formation. Yet, how SG biogenesis contributes to cell survival after lysosomal damage, remains unclear for now.

Major Points:

1. Figure S1B: The number of poly-mRNA (Cy3-oligo-dt) dots appears to be rather low in the images, but the quantification indicates that cells have around 15 poly-mRNA dots. Please clarify this discrepancy.
2. G3BP1/2 and Lysosome Count: Does the loss of G3BP1/2 affect the overall number of lysosomes in cells?
3. mTORC1 Activity Post Lysosomal Damage: Lysosomal damage induces the loss of mTORC1 activity towards 4EBP, due to the inactivation of RagB. What happens to the activation of other mTORC1 targets (Ulk1, S6K, TFEB, etc.)?
4. Rescue Experiments: Please provide rescue experiments for all knockdown and knockout experiments.
5. SG Formation Timing: The authors demonstrate (e.g., in Figure 1A) that G3BP1/2 DKO cells or eIF2 α KD cells undergo cell death already after 30 minutes of LLOMe treatments. Hence, it seems that SG formation acts very early to ensure cell survival. I'm wondering if conducting the APEX-eIF2 experiment at an earlier timepoint would provide additional insight.
6. Figure 4A Quality: The quality of the images provided in Figure 4A is rather poor. Please provide better images. Does the loss of Alix affect the overall number of lysosomes?
7. Direct Interaction Among Proteins: The authors claim direct protein-protein interactions among ALIX, ALG2, PKR, and PACT. While this conclusion is implied by IP experiments, direct interactions among these proteins have not been formally demonstrated (e.g., using in vitro assays).
8. Figure 6C Quality: The quality of the data provided in Figure 6C must be improved.

Minor Points:

- Figure 1F: Size bars are missing.

Referee comments:

Referee #1: Duran and colleagues investigate the pathway leading to activation of the integrated stress response to lysosomal damage, and the role of the stress granules that result. The authors build on their previous work in which they identified the phosphorylation of the translation initiation factor eIF2alpha by the stress kinase PKR and resulting translational repression upon treatment with lysosome-damaging agents. Here the authors elucidate the molecular details of PKR recruitment to damaged lysosomes via ALIX, an ESCORT machinery protein, and identify PACT as a positive regulator of PKR activity under these conditions. The precise mechanism by which PKR is activated remains to be elucidated. The results also indicate that stress granules, which form as a consequence of translational repression when lysosomes are damaged, contribute to improved cell survival. This has also been observed in the context of various pathogen infections.

The experiments are convincing and carefully controlled. Although the high-content microscopy images are quantified, the absence of a quantitative graph for all blots makes it difficult to visually understand the meaning and validity of the claims. Overall, with many novel findings, this work makes an important contribution to understanding the role of the integrated stress response and stress granules in cell survival, as well as the complex interplay between integrated stress responses and other damage response pathways. The main comments below should be addressed experimentally to reinforce certain conclusions.

We thank the referee for the accurate summary of our findings and constructive suggestions. We have conducted a series of new experiments to address the four major concerns outlined below. We have clarified how PKR is activated upon lysosomal damage, as discussed in question 1 below. We have also added quantitative graphs for all blots (including Figs. 2B, 2C, 3C, 3E, 3F (EV3B), 4B, 4D, 4E, 4F, 5A, 5C, 5E-G, 6B-F, as well as new data, Figs. 5D, EV2H-J). These enhancements significantly reinforce the conclusions of our study.

1. The missing piece of evidence concerns the triggering of PKR activation. Does PKR detect cellular dsRNA? Is there an accumulation of dsRNA in the cytoplasm after LLOMe treatment, e.g. detectable by immunofluorescence using the J2 antibody? One way of answering this question in part is, as done in Figure 3 for PKR wt, to knockdown PKR and reconstitute cells with a PKR dsRNA-binding mutant (e.g. K60A and K150A, described in PMID: 7852324 and PMID: 8810342).

We appreciate the referee's comments, and as recommended, we have carried out four new sets of experiments to address these issues:

(i) We tested dsRNA accumulation using the J2 antibody in response to LLOMe treatment. We did not observe an increase in dsRNA/J2 puncta number after LLOMe treatment, a condition that can induce PKR activation. We used poly(I:C) treatment as a positive control (**Display 1**, new data, Fig. EV2G; text on p. 7).

Display 1, new data, Fig. EV2G. There is no dsRNA accumulation in the cytoplasm during LLOMe treatment.

(ii) We previously knocked down the lysosomal RNase RNASET2 but did not detect a change in stress granule formation in response to lysosomal damage (Jia et al., 2022; PMID: 36179369). In our new experiment, we tested PKR activation in RNASET2 knockdown cells and did not observe a change upon LLOMe treatment (**Display 2**, new data, Fig. EV2H; text on p. 7).

Display 2, new data, Fig. EV2H. Lysosomal RNase does not affect PKR activation upon

(iii) As recommended, we generated a PKR mutant deficient in dsRNA-binding ability (PKR^{K60A&K150A}). We reconstituted PKR knockout cells with GFP-PKR and GFP-PKR^{K60A&K150A}, both of which restored PKR activation triggered by LLOMe (**Display 3**, new data, Fig. EV2I; text on p. 7).

Display 3, new data, Fig. EV2I. PKR activation during lysosomal damage occurs independently of its dsRNA binding ability.

(iv) We showed that PACT acts as a positive endogenous regulator of PKR activation in response to lysosomal damage through knockdown experiments (Fig. 3E). To further strengthen this conclusion, we reconstituted PACT knockdown cells with PACT and observed the restoration of PKR activation (**Display 4**, new data, Fig. EV2J; text on p. 7).

Display 4, new data, Fig. EV2J. PKR activation during lysosomal damage is dependent on its endogenous activator PACT.

In summary, all evidence suggests that PKR activation in response to lysosomal damage is triggered not by dsRNA accumulation, but by its endogenous activator PACT, as supported by Figures 3, EV2 and EV3. Numerous studies report that PACT promotes PKR dimerization and the conformational changes necessary for activation through direct binding. Furthermore, we found that lysosomal damage triggers an increased association between PKR and PACT, with ALIX and galectin-3 contributing to this interaction, as supported by Figures 4, 5, 6, EV4 and EV5. We have further elaborated on the mechanisms of PKR activation upon lysosomal damage in the discussion on p. 12.

2. The authors use G3BP1/2 double KO cells to measure the impact of stress granules on cell survival during LLOMe treatment and observe higher survival in cells unable to form SGs. A similar result was obtained in the presence of CHX or ISRIB. However, although useful, these approaches remain indirect. In recent years, it has been shown that G3BP1 is a multifunctional protein, also involved in ribosome quality control, PKR regulation, etc. On the other hand, the cycloheximide and ISRIB interfere at different levels with translation. Direct proof could be provided by the use of G3BP small-molecule inhibitors FAZ3532 or FAZ3780 recently published by the Taylor laboratory (PMID: 38284934) and now commercially available. In an identical context of translation inhibition and in the presence of G3BP1, they would demonstrate that stress granule assembly itself is necessary for cell survival.

We thank the referee for this constructive suggestion. As recommended, we employed FAZ3532 and FAZ3780 to inhibit stress granule formation and test cell survival during lysosomal damage. Our findings show that: (1) these drugs individually abolish stress granule formation upon LLOMe treatment (**Display 5**, new data, Fig. 1H). (2) Stress granule-deficient cells induced by these inhibitors exhibit significantly increased cell death, as demonstrated by PI uptake assay and LDH release assay (**Display 6**, new data, Fig. 1I and 1J). Thus, these data emphasize that stress granule assembly itself is necessary for cell survival during lysosomal damage (text on p. 5).

Display 5, new data, Fig. 1H. FAZ3532 or FAZ3780 inhibits stress granule formation in response to lysosomal damage.

Display 6, new data, Fig. 1I and 1J. Stress granule deficiency induced by FAZ3532 or FAZ3780 increases cell death during lysosomal damage.

3. Figure 6B: p-PKR is not visibly detectable in the LLOMe-treated U2OS control (SCR), whereas it is in Figures 3E and 4B. I think the choice of blot is not the best. The same applies to figure 6C. IP staining for PACT looks "dubious" due to the absence of staining in lanes 2 to 4. A better choice of images would avoid this impression, but overall, as mentioned above, quantification of all blots is essential.

We thank the referee for pointing out this issue. As suggested, we have provided an improved set of blots for Figure 6B, along with quantification for p-PKR and p-eIF2 α . Additionally, we have provided an improved set of blots for Figure 6C, with quantification for the interaction between Gal3 and ALIX, PKR and PACT (**Display 7 and 8**, new Fig. 6B and 6C).

As requested, we have added quantitative graphs for all blots including Figs. 2B, 2C, 3C, 3E, 3F (EV3B), 4B, 4D, 4E, 4F, 5A, 5C, 5E-G, 6B-F, and new data in Figs. 5D, EV2H-J.

Display 7, new Fig. 6B. Galectin-3 (Gal3) negatively regulates the phosphorylation of PKR and eIF2 α in response to lysosomal damage.

Display 8, new Fig. 6C. Galectin-3 (Gal3) associates with ALIX, PKR and PACT in response to lysosomal damage.

4. Figure 7 is very interesting, linking lysosomal damage to contexts more relevant than that of LLOMe use. However, the conclusions are somewhat preliminary at this stage, as none of the results provide information on the involvement of PKR and PACT in these processes. Is PKR phosphorylated during these different treatments? Figure 7B and S6D show only eIF2 α phosphorylation. Can these results on survival in the various pathological situations be reversed by the C16 inhibitor?

We appreciate the referee's constructive suggestion. Following the recommendation, we tested the phosphorylation of PKR in samples from Figure 7B and Appendix Figure S1D. We observed corresponding activation of PKR along with eIF2 α phosphorylation (**Display 9 and 10**, new Fig. 7B and Appendix Fig. S1D; text on p. 11).

Display 9, new Fig. 7B. Lysosomal damage induced by human adenovirus infection initiates the phosphorylation of PKR and eIF2 α .

Display 10, new Appendix Fig. S1D. Various disease agents that cause lysosomal damage can initiate the phosphorylation of PKR and eIF2 α .

Additionally, as suggested, we applied the PKR inhibitor C16 to assess cell death using an LDH release assay under various pathological conditions that can induce lysosomal damage. We observed that C16 treatment increased cell death under all testing conditions, highlighting the role of the PKR-eIF2 α pathway in controlling stress granule formation and promoting cell survival in disease contexts (**Display 11**, new data, Appendix Figs. S1G-K; text on pp. 11-12).

Display 11, new data, Appendix Figs. S1G-K. The PKR inhibitor C16 exacerbates cell death in response to lysosomal damage triggered by various pathological conditions.

Referee #2: This manuscript describes "a mechanism that transduces lysosomal damage signals to stress granule formation for cell survival." The authors use a combination of cell biological and biochemical methods to demonstrate that ALIX and ALG2 sense calcium leakage from damaged lysosomes. This leads to the activation of ALIX and the association of PKR with its activator PACT. Activated PKR phosphorylates eIF2 α , which in turn leads to the G3BP1/2-dependent formation of stress granules (SGs). The activation of this pathway, induced by lysosomal damage and the formation of SGs, is important for cell survival in response to diverse pathogenic challenges (hemozoin, adenoviral infections) associated with major human diseases. The authors delineate (supported by data) the pathway leading from lysosomal damage to SG formation. Yet, how SG biogenesis contributes to cell survival after lysosomal damage, remains unclear for now.

We thank the referee for the accurate summary of our study and for the valuable suggestions provided. In response, we have conducted a series of experiments to address the major points and clarify any weaknesses. This has resulted in 18 new and revised panel data (Figs. 1F, 2B, 4A, 5D, 6C, EV1A, EV1B, EV1H, EV2A, EV2D, EV2E, EV2J, EV4A, EV4E, EV4F, EV5C, EV5D and EV5G). In addition, we agree with the referee that how SG biogenesis contributes to cell survival after lysosomal damage remains unclear for now. Recently, a study suggested that stress granules can plug holes of damaged lysosomes and stabilize damaged lysosomal membranes (PMID: 37968398). However, many open questions still exist regarding the role of stress granules in contributing to cell survival upon lysosomal damage, which is a key focus for our future research. We have emphasized this topic in the discussion section on p. 13-14. Overall, we believe these changes significantly strengthen our study.

Major Points:

1. Figure S1B: The number of poly-mRNA (Cy3-oligo-dt) dots appears to be rather low in the images, but the quantification indicates that cells have around 15 poly-mRNA dots. Please clarify this discrepancy.

We thank the referee for pointing out this issue and apologize for the oversight. The data and graphs are derived from a large image database generated using high-content microscopy, which is then machine-processed to produce the graphs and accompanying statistics. We have now selected images from this database that more accurately represent the data shown in the bar graphs. Please see the new S1B (EV1B(i)) for these updated images.

2. G3BP1/2 and Lysosome Count:
Does the loss of G3BP1/2 affect the overall number of lysosomes in cells?

We thank the referee for this comment. As recommended, we conducted two experiments: (1) we examined the expression level of LAMP2 (a lysosomal marker) in samples from wildtype and G3BP1/2 knockout cells and observed no change (Display 12, new Fig. EV1A); (2) We assessed the overall number of lysosomes by staining for LAMP2 in wildtype and G3BP1/2 knockout cells using high-content microscopy. We observed no significant changes at the resting state or during lysosomal damage (Display 12, new data, Fig. EV1B(ii); text on p. 4).

Display 12, new Fig. EV1A and new data, Fig. EV1B(ii). Depletion of G3BP1 and G3BP2 does not affect the overall number of lysosomes in cells.

3. mTORC1 Activity Post Lysosomal Damage: Lysosomal damage induces the loss of mTORC1 activity towards 4EBP, due to the inactivation of RagB. What happens to the activation of other mTORC1 targets (UlK1, S6K, TFEB, etc.)?

We appreciate the referee's comment. As recommended, we have tested other mTORC1 targets in the samples, including S6K(Thr389), ULK1 (Ser757) and TFEB (Ser142). These targets showed the same reduction in phosphorylation as 4EBP1 (Ser65) in response to lysosomal damage. This is consistent with our previous publications (Jia et al., 2018, PMID: 29625033; Jia et al., 2022, PMID: 36179369), which also report reduced phosphorylation levels of 4EBP1 (Ser65), S6K(Thr389), ULK1 (Ser757) and TFEB (Ser142) upon lysosomal damage. In addition, the inactivation of mTORC1 upon lysosomal damage has been reported and replicated by other groups (PMID: 31432621 and 27725083), who detected decreased phosphorylation of mTORC1 downstream targets as well as mTOR dissociation from damaged lysosomes. We have included these new data in Figs. EV2A and 2B (**Display 13**, text on pp. 5-6).

Display 13, new Fig. EV2A and Fig. 2B. Lysosomal damage leads to reduced mTORC1 activity, as evidenced by decreased phosphorylation of its targets 4EBP1, S6K, ULK1, and TFEB.

4. Rescue Experiments: Please provide rescue experiments for all knockdown and knockout experiments.

We thank the referee for this valuable suggestion. In response, we have performed rescue experiments for knockdown and knockout studies as requested. The results, including new data (Figs. EV1H, EV2I, EV2J, EV4E, EV4F, EV5G) and previously obtained data, have been incorporated into our manuscript, providing further validation of our findings. The details are as follows:

- New data Fig. EV1H: G3BP1/2 rescue in G3BP knockout cells results in increased cell survival upon lysosomal damage (text on p. 5).
- Fig. 2E: eIF2α rescue in eIF2α knockdown cells results in increased stress granule formation upon lysosomal damage (text on p. 6).
- Fig. 3C and new data Figs. EV2I and EV2J: PKR and PACT rescue in their respective deficient cells results in increased eIF2α or PKR activation upon lysosomal damage (text on p. 7).
- New data Figs. EV4E and EV4F: ALIX and ALG2 rescue in their respective deficient cells results in increased eIF2α activation and stress granule formation upon lysosomal damage (text on p. 8).

- Figs. 5E and 5F, and new data Fig. EV5G: ALIX rescue, controlling over the interaction between PACT and PKR, results in increased activation of PKR on damaged lysosomes (text on p. 9).
- Figs. 6D-F reflect galectin-3 rescue, demonstrating its control over the interaction among ALIX, PACT and PKR, and the activation of PKR upon lysosomal damage (text on p. 10).

5. SG Formation Timing: The authors demonstrate (e.g., in Figure 1A) that G3BP1/2 DKO cells or eIF2 α KD cells undergo cell death already after 30 minutes of LLOMe treatments. Hence, it seems that SG formation acts very early to ensure cell survival. I'm wondering if conducting the APEX-eIF2 experiment at an earlier timepoint would provide additional insight.

We appreciate the referee's insightful comment. In response, we conducted kinetic experiments to determine the early timing of eIF2 α phosphorylation upon LLOMe treatment. These experiments were performed in two cell lines: (1) HEK293T cells expressing APEX2-eIF2 α , which were used for proteomic analysis of eIF2 α interactomes; (2) U2OS cells, the primary cell line used throughout this study. We found that: in HEK293T cells expressing APEX2-eIF2 α , 1mM LLOMe treatment for 1h initiated eIF2 α phosphorylation, while no significant increase in cell death was observed at this timepoint; In U2OS cells, 2 mM LLOMe treatment for 30 min initiated eIF2 α phosphorylation (**Display 14**, new data, Figs. EV2D and 2E, text on p. 6) and as the referee pointed out, cells began to undergo cell death at this timepoint.

Display 14. new data, Figs. EV2D and EV2E. Kinetics of eIF2 α phosphorylation in HEK293T and U2OS cells in response to lysosomal damage.

Our observations suggest that eIF2 α phosphorylation in response to lysosomal damage is both dose- and time-dependent. Importantly, the proteomic analysis of APEX2-eIF2 α performed in HEK293T cells (1mM LLOMe for 1h) reflects early events of eIF2 α phosphorylation before the onset of significant cell death. We agree with the referee that conducting APEX2-eIF2 α experiments at an earlier timepoint would provide additional insights for further investigation. Based on our kinetic results, we predict that eIF2 α phosphorylation in HEK293T cells may occur between 30min and 1h treatment with 1mM LLOMe. Nevertheless, we believe the signaling pathway proposed in this study reflects both the early events of eIF2 α phosphorylation as well as stress granule formation in response to lysosomal damage.

6. Figure 4A Quality: The quality of the images provided in Figure 4A is rather poor. Please provide better images. Does the loss of Alix affect the overall number of lysosomes?

We thank the referee for pointing out this issue. As mentioned in our response to question 1, we have replaced the previous images with higher-quality versions from the database to improve image quality (**Display 15**, new Fig. 4A). Please see the new Fig. 4A for these updated images.

Furthermore, we examined the effect of ALIX knockdown on the overall number of lysosomes. We observed no significant changes in the number of LAMP2 puncta (a lysosomal marker) in ALIX knockdown cells (**Display 16**, new data, Fig. EV4A, text on p. 8). This aligns with our previous findings that ALIX depletion does not affect lysosomal function, as demonstrated by lysosomal acidification using the LysoTracker assay and cathepsin B activity using the Magic Red assay (Jia et al., 2020, PMID: 31813797, shown in Figs. 1G and S1K).

Display 15, new Fig. 4A. ALIX depletion impairs stress granule formation in response to lysosomal damage.

Display 16, new data, Fig. EV4A. ALIX depletion does not affect the overall number of lysosomes in cells.

7. Direct Interaction Among Proteins: The authors claim direct protein-protein interactions among ALIX, ALG2, PKR, and PACT. While this conclusion is implied by IP experiments, direct interactions among these proteins have not been formally demonstrated (e.g., using in vitro assays).

We appreciate the referee's comment. As suggested, we performed in vitro GST pull-down assays for these proteins. Considering the structural changes of ALIX and ALG2 triggered by calcium, as implied in CoIP assays (such as Fig. 5C, upon lysosomal damage-induced calcium release) and prior research, we treated GST-tagged ALIX or ALG2 with CaCl₂ for the GST pull down assay. We found that ALIX or its partner ALG2 individually did not directly interact with PACT or PKR (**Display 17**, new data, Figs. EV5C and EV5D).

Display 17, new data, Fig. EV5C and EV5D. GST pull-down assays revealed that ALIX or ALG2 individually did not directly interact with PACT or PKR.

However, ALIX and ALG2 together were able to directly interact with the PACT-PKR complex (**Display 18**, new data, Fig. 5D). These results suggest that conformational changes are important for this direct interaction, possibly induced by ALG2 exposing the V domain of ALIX and PACT promoting PKR dimerization (text on p. 9).

Display 18, new data, Fig. 5D. GST pull-down assays demonstrated that ALIX and ALG2 together directly interacted with the PACT-PKR complex.

8. Figure 6C Quality: The quality of the data provided in Figure 6C must be improved.

We thank the referee for pointing out this issue. As suggested by both referees, we have provided an improved set of blots for Figure 6C, along with quantification for the interaction between galectin-3 (Gal3) with ALIX, PKR and PACT (**Display 19**, new Fig. 6C).

Display 19, new Fig. 6C, repeat of display 8. Galectin-3 (Gal3) interacts with ALIX, PKR and PACT in response to lysosomal damage.

Minor Points:

- Figure 1F: Size bars are missing.

Thank you for your suggestion. As requested, we have added size bars to Figure 1F.

Dear Dr. Jingyue,

I am pleased to inform you that your manuscript has been accepted for publication in the EMBO Journal.

Congratulations to you and your team!

COuld you please provide me (in the body of an email is fine) with a two-sentence summary and 3-5 bullet points covering the article's highlights? This will be for the table of contents on our website.

Best wishes,

William

William Teale, PhD
Editor
The EMBO Journal
w.teale@embojournal.org

Referee #1:

Duran and colleagues provided comprehensive answers to my questions and addressed all my concerns as well as those of the second reviewer. Through this additional work, they have provided several insights that clarify the mechanism of PKR activation in response to LLOMe and demonstrate the importance of stress granules in cell survival under these conditions. Altogether, this work makes an important contribution to our understanding of the integrated stress response and crosstalk with lysosomal damage.

Referee #2:

The authors did a great job during the revision and have addressed the majority of my concerns.
